# p97/VCP is required for piecemeal autophagy of aggresomes

Maria Körner[1,5], Paul Müller [●][1,5], Hirak Das [●][2], Felix Kraus[3], Timo Pfeuffer [●][1], Sven Spielhaupter [●][1], Silke Oeljeklaus [●][2], Christina Schülein-Völk[4], J. Wade Harper [●][3], Bettina Warscheid [●][2] & Alexander Buchberger [●][1] [✉]

Metazoan cells adapt to the exhaustion of protein quality control (PQC) systems by sequestering aggregation-prone proteins in large, pericentriolar structures termed aggresomes. Defects in both aggresome formation and clearance affect proteostasis and have been linked to neurodegenerative diseases, but aggresome clearance pathways are still underexplored. Here we show that aggresomes comprising endogenous proteins are cleared via selective autophagy requiring the cargo receptor TAX1BP1. TAX1BP1 proximitomes reveal the presence of various PQC systems at aggresomes, including Hsp70 chaperones, the 26S proteasome, and the ubiquitin-selective unfoldase p97/VCP. While Hsp70 and p97/VCP with its cofactors UFD1-NPL4 and FAF1 play key roles in aggresome disassembly, the 26S proteasome is dispensable. We identify aggresomal client proteins that are degraded via different routes, in part in a p97/VCP-dependent manner via aggrephagy. Upon acute inhibition of p97/VCP, aggresomes fail to disintegrate and cannot be incorporated into autophagosomes despite the presence of factors critical for aggrephagosome formation, including p62/SQSTM1, TAX1BP1, and WIPI2. We conclude that the p97/VCP-mediated removal of ubiquitylated aggresomal clients is essential for the disintegration and subsequent piecemeal autophagy of aggresomes.

Maintaining the proper conformation and function of proteins is essential for cellular homeostasis. Under various physiological and pathological conditions, proteins can misfold and form aggregates, which are implicated in neurodegenerative diseases such as Alzheimer's, Parkinson's, and Huntington's diseases, as well as in cancer and other disorders[1–3]. Consequently, cells have evolved intricate protein quality control (PQC) systems, including the ubiquitin-proteasome system and molecular chaperones such as the heat-shock protein 70 (Hsp70), to prevent or reduce protein aggregation[4,5]. Additionally, they utilize selective macroautophagy (hereafter autophagy) pathways to remove aggregated proteins and maintain proteostasis[6,7]. Another prominent defense strategy is the formation of distinct inclusions, wherein protein aggregates are sequestered, concentrated, and stored for subsequent solubilization and/or degradation[8].

Aggresomes are large, perinuclear, vimentin-encaged inclusions formed by metazoan cells through the active transport of ubiquity-lated aggregates along microtubules to the centrosome[9,10]. They suppress the detrimental effects of dispersed smaller aggregates and are therefore considered cytoprotective[11–13]. Defects in both aggresome formation and post-stress clearance affect cellular proteostasis and have been linked to neurodegenerative diseases[14]. While the formation of aggresomes has been studied in considerable detail (reviewed in ref. 15), their protein content and clearance pathways are incompletely characterized.

[1]Biocenter, Chair of Biochemistry I, University of Würzburg, Würzburg, Germany. [2]Biocenter, Chair of Biochemistry II, University of Würzburg, Würzburg, Germany. [3]Department of Cell Biology, Harvard Medical School, Boston, MA, USA. [4]Biocenter, Core Unit High-Content Microscopy, University of Würzburg, Würzburg, Germany. [5]These authors contributed equally: Maria Körner, Paul Müller. [✉]e-mail: alexander.buchberger@uni-wuerzburg.de

The clearance of aggresomes and large aggregates formed upon overexpression of disease-linked, aggregation-prone proteins, or upon seeding with aggregates thereof, has been variably reported to occur via autophagy[16–21] or proteasomal degradation[22–24] and to involve the ubiquitin-selective unfoldase and disaggregase p97 (also known as VCP)[24–27]. However, the chosen experimental systems typically involved artificially high expression levels and lacked temporal resolution of aggresome formation and clearance due to ongoing ectopic expression, thus complicating their interpretation with respect to clearance pathways. Alternatively, aggresomes consisting of endogenous proteins have been shown to be induced by pharmacological inhibition of the 26S proteasome in various cell lines and primary rat neurons[28–30]. Moreover, in attempts to establish mouse models of Lewy body dementia, the local genetic depletion or pharmacological inhibition of the 26S proteasome in mouse brain was shown to induce the formation of intraneuronal, Lewy body-like proteinaceous inclusions[31,32], linking aggregate formation upon proteasomal impairment to neurodegeneration. Importantly, Lewy bodies share a number of features with aggresomes and have been proposed to resemble failed, persistent aggresomes[33]. Clearance of aggresomes comprising endogenous proteins was reported to proceed via autophagy[34,35] with the help of p97 (refs. [25,36,37]) and the proteasomal de-ubiquitylating enzyme PSMD14 (also known as RPN11, Poh1), which generates free ubiquitin chains that in turn activate the deacetylase HDAC6 for cortactin-dependent F-actin remodeling[35]. However, the protein content of such endogenous aggresomes is unknown, and the precise mechanisms and order of events underlying their clearance are still poorly understood.

Here, we demonstrate that aggresomes consisting of endogenous proteins are broken up into smaller fragments and subsequently cleared by aggrephagy in a process requiring p97 and its cofactors UFD1-NPL4 and FAF1, Hsp70 and its co-chaperone DNAJB6, as well as the selective autophagy receptor (SAR) TAX1BP1. We identify endogenous aggresomal proteins and show that some are degraded via proteasomes or aggrephagy in a p97-dependent manner. We reveal that aggresomes fail to disintegrate and to become incorporated into aggrephagosomes in the presence of p97 inhibitors, indicating that the critical function of p97 in the clearance of aggresomes is their fragmentation.

## Results

### Aggresomes of endogenous proteins are cleared by selective autophagy

To characterize the turnover of aggresomes formed by endogenous proteins, we established conditions allowing us to follow aggresome formation and clearance in HeLa cells (Fig. 1a, b). After an 8 h treatment with the proteasome inhibitor Bortezomib (Btz), about 15% of the cells contained ubiquitin- (Ub-) positive, vimentin-encaged perinuclear aggresomes. This number increased to around 40% after 15 h of recovery (i.e., 15 h after washout of Btz), by which time proteasome activity was fully restored (Supplementary Fig. 1a), and subsequently declined to pre-stress levels between 22 and 36 h of recovery. Treatment with the chemically distinct proteasome inhibitors MG-132 and Epoxomicin similarly induced aggresome formation and clearance, albeit with slightly different kinetics (Supplementary Fig. 1b, c). Aggresome formation and clearance were also observed upon permanent treatment with Btz, MG-132, and Epoxomicin (Supplementary Fig. 1d), suggesting that inhibitor washout is not strictly required for aggresome clearance. However, to allow for a more precise control of the start of aggresome clearance and to avoid the toxic effects of prolonged inhibitor treatment, we used the washout protocol in all further experiments. Aggresome clearance proceeded via autophagy, as it coincided with a strong increase in lipidated LC3 (Fig. 1c), involved decoration of ubiquitylated material with LC3 in an ATG7-dependent manner (Fig. 1d), and was impaired in *ATG7* knockout cells (Fig. 1e, f). Moreover, addition of the inhibitor of autophagosome–lysosome fusion and lysosomal degradation, Bafilomycin A1 (Baf), induced the

accumulation of Ub- and LC3-positive, bona fide aggrephagosomes and autolysosomes (Fig. 1g). Testing various SARs, we observed a strong decoration of early, peripheral aggregates and of mature, perinuclear aggresomes with p62/SQSTM1 and TAX1BP1, as well as a partial colocalization of NBR1, NDP52 and OPTN with aggregates and aggresomes (Fig. 1h). TAX1BP1 is involved in xenophagy[38], mitophagy[39,40], as well as in the aggrephagy of peripheral, puromycin-induced aggregates, ectopically expressed polyglutamine (polyQ) inclusions, and p62-containing condensates[41,42]. It robustly co-localized with Ub-positive aggregates/aggresomes throughout the time course of the experiment, well ahead of LC3 (Supplementary Fig. 2a–c), and was required for efficient aggresome clearance (Fig. 1i, j; Supplementary Fig. 2d–g), making it an ideal marker to study aggresome biology. Together, these results show that aggresomes formed upon proteasomal inhibition by endogenous proteins are cleared via selective autophagy involving the SAR TAX1BP1.

### Various protein quality control systems are present at the aggresomes

To explore the composition of aggresomes and aggrephagosomes comprising endogenous protein aggregates, we performed proximity proteomics using a knock-in cell line expressing a fusion of the TurboID biotin ligase[43] with TAX1BP1 at endogenous levels (Supplementary Fig. 3a, b). Immunofluorescence microscopy confirmed that TurboID-TAX1BP1 localized to aggresomes and catalyzed the biotinylation of aggresomal material (Fig. 2a, Supplementary Fig. 3c). Affinity purification of biotinylated proteins followed by label-free quantitative mass spectrometry revealed that the TurboID-TAX1BP1 proximitome under aggrephagy conditions (i.e., 8 h Btz, followed by 15 h recovery and 3 h Baf), compared to control cells expressing free TurboID, was enriched in autophagy-related proteins, the 26S proteasome, molecular chaperones, p97, and miscellaneous proteins such as AMOTL2, PLEK2, FAM83D and PFN2 (Fig. 2b, c; Supplementary Fig. 3d; Supplementary Data 1 & 2). Comparing the TurboID-TAX1BP1 proximitomes under non-stress versus aggrephagy conditions again showed a strong enrichment of chaperones and the 26S proteasome, and also identified p97, AMOTL2, PLEK2, FAM83D, and PFN2 (Fig. 2b, d; Supplementary Fig. 3e, f; Supplementary Data 1 & 2). The aggresome association of several hits, including 19S and 20S proteasomal subunits, p97, Hsp70 and DNAJB6, the autophagy factors TBK1, p62 and TNIP1, as well as AMOTL2, PLEK2, FAM83D and PFN2, was validated by immunofluorescence microscopy and/or immunoblots of the affinity-purified biotinylated material (Fig. 2e, Supplementary Fig. 3g, h). To our knowledge, this is the first description of proteins associated with endogenous aggresomes and aggresome-derived aggrephagosomes.

### The p97 and Hsp70 disaggregases mediate aggresome disintegration

The presence of different PQC systems at aggresomes prompted the question about their respective roles in aggresome clearance. We reasoned that genetic depletion of key PQC factors perturbs general proteostasis and likely affects the formation and composition of aggresomes, thereby confounding the analysis of aggresome clearance. Instead, we added pharmacological inhibitors of PQC systems after 15 h of recovery for 6 h to specifically address their function during the clearance of pre-formed aggresomes. Intriguingly, the p97 inhibitor CB-5083 (CB) completely blocked aggresome clearance, as quantified by the number of cells retaining a large, perinuclear aggresome, whereas Btz and Baf had only minor effects, and the Hsp70 inhibitor VER-155008 (VER) had an intermediate effect (Fig. 3a, b). Quantification of the total aggregate area per cell showed a strong accumulation for CB and VER, but also significant increases for Btz and Baf, indicating the accumulation of smaller, non-aggresomal aggregates in the presence of the latter two inhibitors (Fig. 3c). The inhibitor

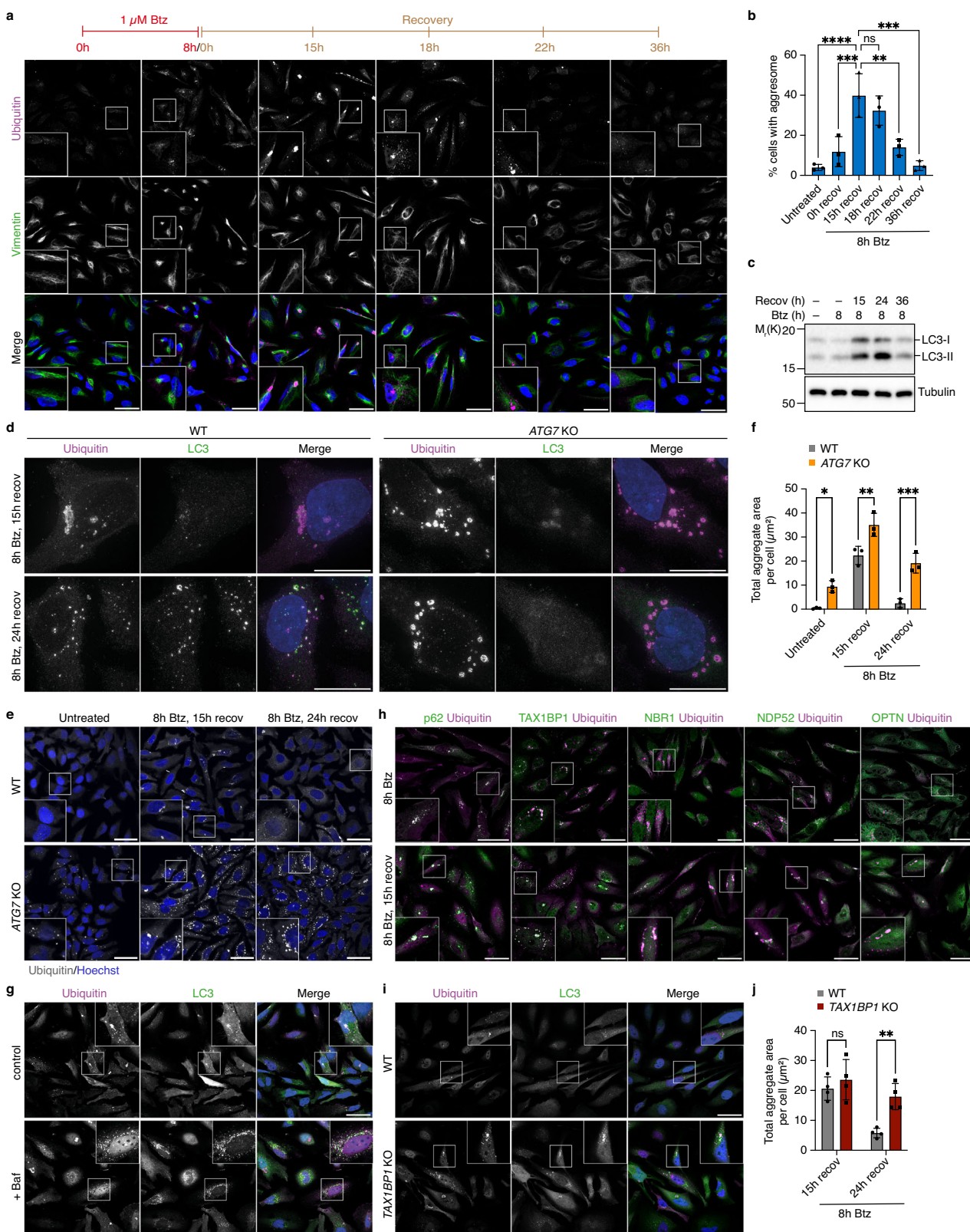

of the PSMD14 deubiquitylase subunit of the 19S proteasomal regulatory complex, Capzimin (Cpz)[44], had the same effect as Btz on aggresome clearance (Supplementary Fig. 4a, b), confirming that the 26S holoproteasome is required for the efficient clearance of smaller aggregates, but not for aggresome disintegration. Immunoblot analysis of RIPA-insoluble proteins revealed that ubiquitylated aggregates present at 15 h of recovery were largely cleared at 21 h under control conditions, but persisted in the presence of any of the PQC inhibitors, in particular in the presence of Btz and CB (Fig. 3d). In control experiments, inhibitor treatment for 6 h without prior Btz treatment did not result in the formation of microscopically detectable aggresomes/aggregates (Supplementary Fig. 4c, d) and induced a moderate amount of RIPA-insoluble ubiquitylated proteins for Btz, but not the other inhibitors (Supplementary Fig. 4e).

**Fig. 1 | Aggresomes comprising endogenous proteins are cleared via selective autophagy. a** HeLa cells were treated with Btz (1 μM, 8 h), recovered for 0 to 36 h, and analyzed by confocal immunofluorescence microscopy using antibodies against ubiquitin and vimentin. Scale bars, 50 μm. **b** Quantification of the percentage of cells with an aggresome in **a**. Shown is the mean ± SD from n = 3 biological replicates with ≥55 cells per time point and replicate; one-way ANOVA. **c** Lysates from HeLa cells that recovered for the indicated times from Btz treatment were analyzed by immunoblot using antibodies against LC3 and tubulin. Representative result from three independent experiments. **d** HeLa$^{TMEM192-3xHA}$ wildtype (WT) and *ATG7* knockout (KO) cells recovered for 15 h or 24 h from Btz treatment were analyzed by 3D structured illumination microscopy (SIM) using antibodies against ubiquitin and LC3B. Maximum intensity projections are shown. Scale bars, 20 μm. Representative result from two independent experiments. **e** HeLa$^{TMEM192-3xHA}$ WT and *ATG7* KO cells left untreated or recovered for 15 h or 24 h from Btz treatment were analyzed by confocal immunofluorescence microscopy using an anti-ubiquitin antibody. Scale bars, 50 μm. **f** Quantification of total aggregate area per cell of the experiment shown in (**e**). Shown is the mean ± SD from n = 3 biological replicates with ≥85 cells per time point and replicate; two-way ANOVA. **g** Cells recovered for 21 h from Btz treatment in the presence or absence of 100 nM Baf for the last 6 h were analyzed by confocal immunofluorescence microscopy using antibodies against ubiquitin and LC3B. Scale bars, 50 μm. Representative result from four independent experiments. **h** Localization of SARs in HeLa cells was analyzed after 8 h Btz treatment ±15 h recovery by confocal immunofluorescence microscopy using the indicated antibodies. Scale bars, 50 μm. Representative result from two independent experiments. **i** WT and *TAX1BP1* KO cells recovered for 24 h from Btz treatment were analyzed by confocal immunofluorescence microscopy using antibodies against ubiquitin and LC3. Scale bars, 50 μm. **j** Quantification of the total aggregate area per cell of the experiment shown in **i**. Shown is the mean ± SD from n = 4 biological replicates with ≥80 cells per time point and replicate; two-way ANOVA.

HDAC6 was previously shown to mediate aggresome disintegration and clearance[35]. While HDAC6 was not present in our TAX1BP1 proximitome datasets, consistent with its disappearance from aggresomes during proteasome inhibitor washout[35], addition of the HDAC6 inhibitor Tubastatin A (TBSA) during the last 6 h of recovery impaired aggresome disintegration by about 50% (Supplementary Fig. 4f). The combined addition of CB and the HDAC6 inhibitors Tubacin and TBSA did not affect aggresome disintegration beyond the effect of CB alone (Supplementary Fig. 4f), but caused a slight increase in the total aggregate area (Supplementary Fig. 4g). Any combination of CB and other inhibitors had the same effect as CB alone on aggresome clearance (Supplementary Fig. 4h–k). Together, the microscopy and fractionation data show that p97 and Hsp70 activities are strictly required to break up aggresomes, whereas the proteasome and autophagy degrade smaller aggregates, and HDAC6 appears to be involved in the turnover of both aggresomes and smaller aggregates.

DNAJB6, an Hsp70 co-chaperone previously implicated in the suppression of polyQ protein and nucleoporin aggregate formation[45,46], was the most strongly enriched J domain protein (JDP) in the TAX1BP1 proximitomes (Fig. 2c, d). Intriguingly, *DNAJB6* knockout cells exhibited severely impaired aggresome clearance, and combining *DNAJB6* deletion with CB treatment had an even stronger effect (Fig. 3e–g).

Altogether, our data show that p97 and the Hsp70–DNAJB6 chaperone axis are the key PQC systems mediating aggresome disintegration.

## Aggresome clearance is p97-dependent in various cell types

To explore the importance of p97-dependent aggresome clearance in other cell types, we used retinoic acid-differentiated SH-SY5Y cells (Supplementary Fig. 5a), a well-established, neuron-like in vitro model for studying aggresomes and other disease-linked protein aggregates[47,48]. Treatment with MG-132 for 16 h, followed by inhibitor washout for 8 h, induced the formation of perinuclear, TAX1BP1-positive aggresomes that were readily cleared during the next 16 h of recovery (Fig. 4a–c). Importantly, p97 localized to these aggresomes (Supplementary Fig. 5b), and CB treatment during recovery caused a virtually complete clearance defect by blocking aggresome disintegration (Fig. 4a–c). Aggresome clearance required autophagy, because Baf treatment during recovery resulted in an increase in total aggregate area (Fig. 4a, b, d) and in the accumulation of LC3/ubiquitin (Fig. 4a, e) and TAX1BP1/ubiquitin (Fig. 4b, f) double-positive aggrephagosomes and autolysosomes.

Similarly, impaired aggresome disintegration upon CB treatment was also observed in non-transformed human ARPE-19 epithelial cells (Fig. 4g–i) as well as in the human A549 lung cancer and murine C2C12 myoblast cell lines (Supplementary Fig. 5c–f), indicating that p97 is generally required for the clearance of endogenous aggresomes in a variety of distinct cell types.

## UFD1-NPL4 and FAF1 are required for efficient aggresome clearance

Specific cellular functions of p97 are mediated by a multitude of cofactors controlling its substrate interactions and subcellular localization[49]. We therefore screened aggresomes for the presence of p97 cofactors by immunofluorescence microscopy and found that they were decorated with UFD1 (also known as UFD1L), NPL4 (also known as NPLOC4), and UBXN1, in agreement with a recent report[37], and additionally with FAF1, PLAA, and HOIP (Fig. 5a). Moreover, the TAX1BP1 proximitomes contained FAF2 (Fig. 2c, d; Supplementary Fig. 3f), and proximitomes of TurboID-UBXN1 and PLAA-TurboID under aggrephagy conditions revealed a significant enrichment of NPL4, VCPIP1, and, mutually, PLAA and UBXN1, among other cofactors (Supplementary Fig. 6a–c). We next determined potential functions of these cofactors in aggresome clearance, which had not been addressed before. Since ectopically expressed TurboID-FAF2 fusion proteins co-localized with and biotinylated ER structures but not aggresomes, we excluded FAF2 from further analyses.

We initially focused on HOIP, the catalytic subunit of the linear ubiquitin chain assembly complex (LUBAC)[50], as it was recently shown to be required for the efficient clearance of various aggregates induced by overexpression of disease-associated proteins[20,26]. To dissect the role of HOIP in the formation and clearance of endogenous aggresomes, we used the specific HOIP inhibitor HOIPIN-8 (ref. 51). Addition of HOIPIN-8 during aggresome formation (i.e., in the 15 h following Btz washout) caused a strong increase in the number of aggresome-containing cells (Fig. 5b), whereas its addition during clearance had no effect (Fig. 5c), indicating that HOIP activity is critical in the early proteotoxic stress response, but not during aggresome clearance.

We next tested the role of the heterodimeric cofactor UFD1-NPL4, which primes ubiquitylated substrates for p97-mediated unfolding[52–54]. siRNA-mediated depletion of UFD1 or NPL4 did not affect the formation of aggresomes, but significantly impaired clearance (Fig. 5d, 21 h recovery; Supplementary Fig. 6d). A similar defect was observed upon addition of bis-(diethyldithiocarbamate)-copper (CuET) at the beginning of the clearance phase, a compound that releases cupric ions to disrupt two zinc finger domains in NPL4 and thereby block p97-UFD1-NPL4 activity[55,56] (Supplementary Fig. 6e), thus confirming that an active p97-UFD1-NPL4 complex is required for efficient aggresome clearance. While we did not observe clearance defects upon depletion of UBXN1 and VCPIP1, depletion of PLAA and, more so, FAF1 significantly impaired clearance (Fig. 5d; Supplementary Fig. 6f, g). UFD1-NPL4 and FAF1 cooperatively mediate the interaction of p97 with ubiquitylated substrates[57–60]. Accordingly, the localization of p97 to aggresomes was significantly reduced upon depletion of UFD1-NPL4 (Fig. 5e; Supplementary Fig. 6h), and the co-depletion of FAF1 led to a further reduction (Supplementary Fig. 6h), while the de(p)letion of PLAA, UBXN1, VCPIP1, and HOIP did not affect the recruitment of p97 to aggresomes (Supplementary Fig. 6h–k).

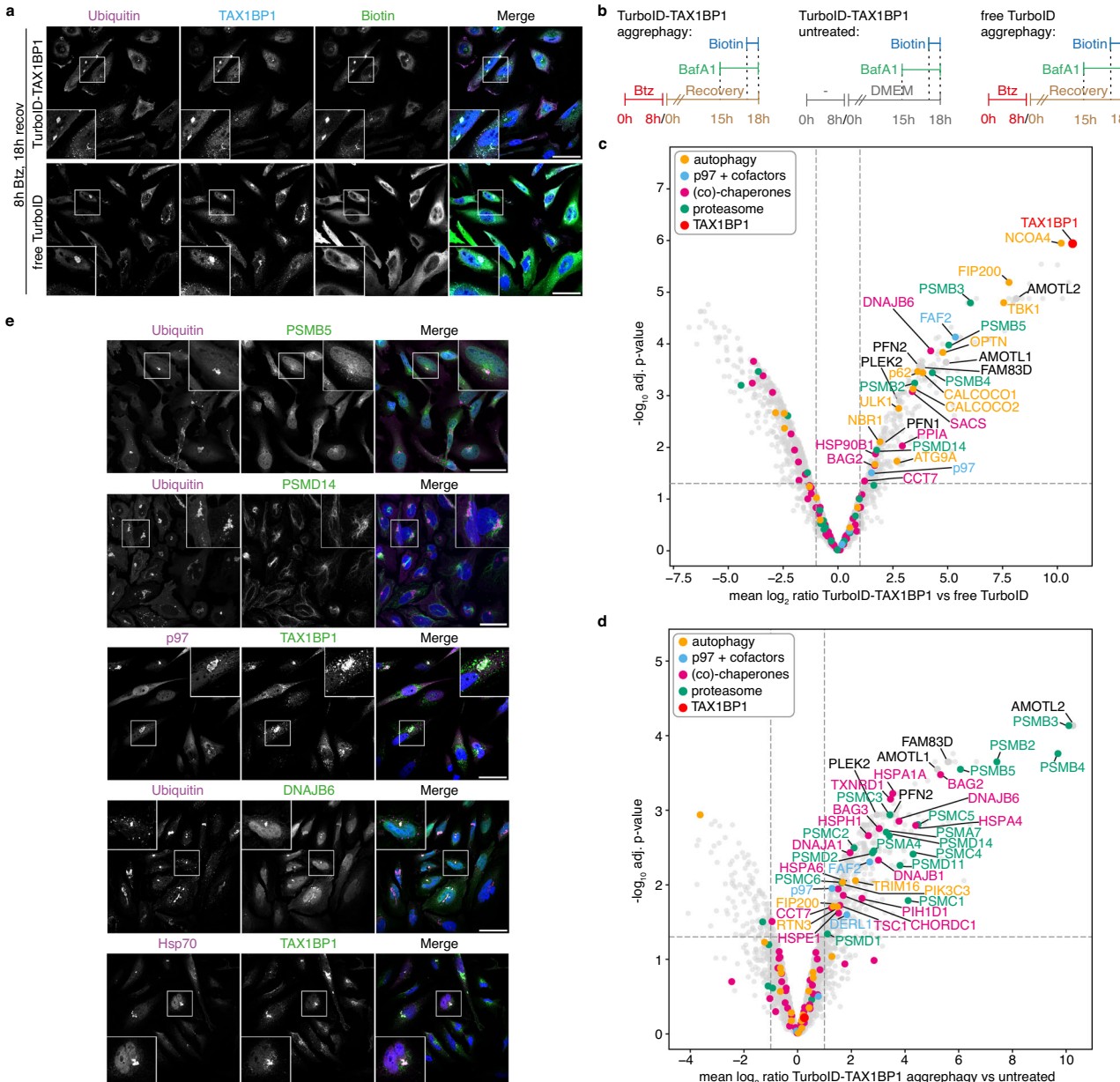

**Fig. 2 | Various protein quality control systems are present at aggresomes.**
**a** Confocal immunofluorescence microscopy of HeLa cells expressing TurboID-TAX1BP1 or free TurboID recovered for 18 h from Btz treatment in the presence of 100 nM Baf for the last 3 h and 50 μM biotin for the last 1 h using the indicated antibodies. Scale bars, 50 μm. Representative result from two independent experiments. **b** Scheme explaining the three experimental conditions used for proximitome analysis. **c, d** Volcano plots for HeLa cells expressing TurboID-TAX1BP1 versus free TurboID under aggrephagy conditions (**c**) or HeLa cells expressing TurboID-TAX1BP1 under aggrephagy versus untreated conditions (**d**), as depicted in **b**. Statistical significance was assessed using the limma package,

employing a two-sided empirical Bayes moderated t-test with false discovery rate (FDR) correction via the Benjamini–Hochberg method. n = 4 biological replicates; thresholds: $\log_2$ FC > ±1, −log10 (adj. p-value) > 1.30 (i.e., adj. p-value < 0.05). Annotation based on gene names (except for p97, p62, FIP200), specific categories of proteins indicated by colored circles. **e** Aggresome localization of highly enriched TAX1BP1 proximity partners was tested in HeLa cells recovered for 18 h from Btz treatment in the presence of 100 nM Baf for the last 3 h by confocal immunofluorescence microscopy using the indicated antibodies. Scale bars, 50 μm. Representative result from two independent experiments.

Interestingly, the impaired recruitment of p97 to aggresomes observed upon knockdown of UFD1-NPL4 correlated with higher nuclear p97 levels (Fig. 5e, f). We therefore tested the effect of depleting VCF1, a recently identified cofactor promoting the nuclear localization of p97 (ref. 61). Depletion of VCF1 caused a reduction of nuclear p97 in control cells, as expected, and restored normal nuclear and aggresomal p97 localizations in NPL4-depleted cells (Fig. 5e–g). Importantly, however, the restored aggresome localization of p97 in cells co-depleted of VCF1 and NPL4 was insufficient to rescue the aggresome clearance defect caused by NPL4 depletion (Fig. 5h).

Taken together, our results show that UFD1-NPL4 and FAF1 are required for the normal recruitment of p97 to aggresomes, and they strongly suggest that both cofactors cooperate in the p97-mediated unfolding or disaggregation of aggresomal proteins as a prerequisite for efficient aggresome clearance.

**Aggresomal client proteins are degraded via different pathways**
Having established the central role of p97 and its cofactors UFD1-NPL4 and FAF1 in aggresome clearance, we sought to identify specific p97 targets at aggresomes. We focused on two proteins, AMOTL2 and

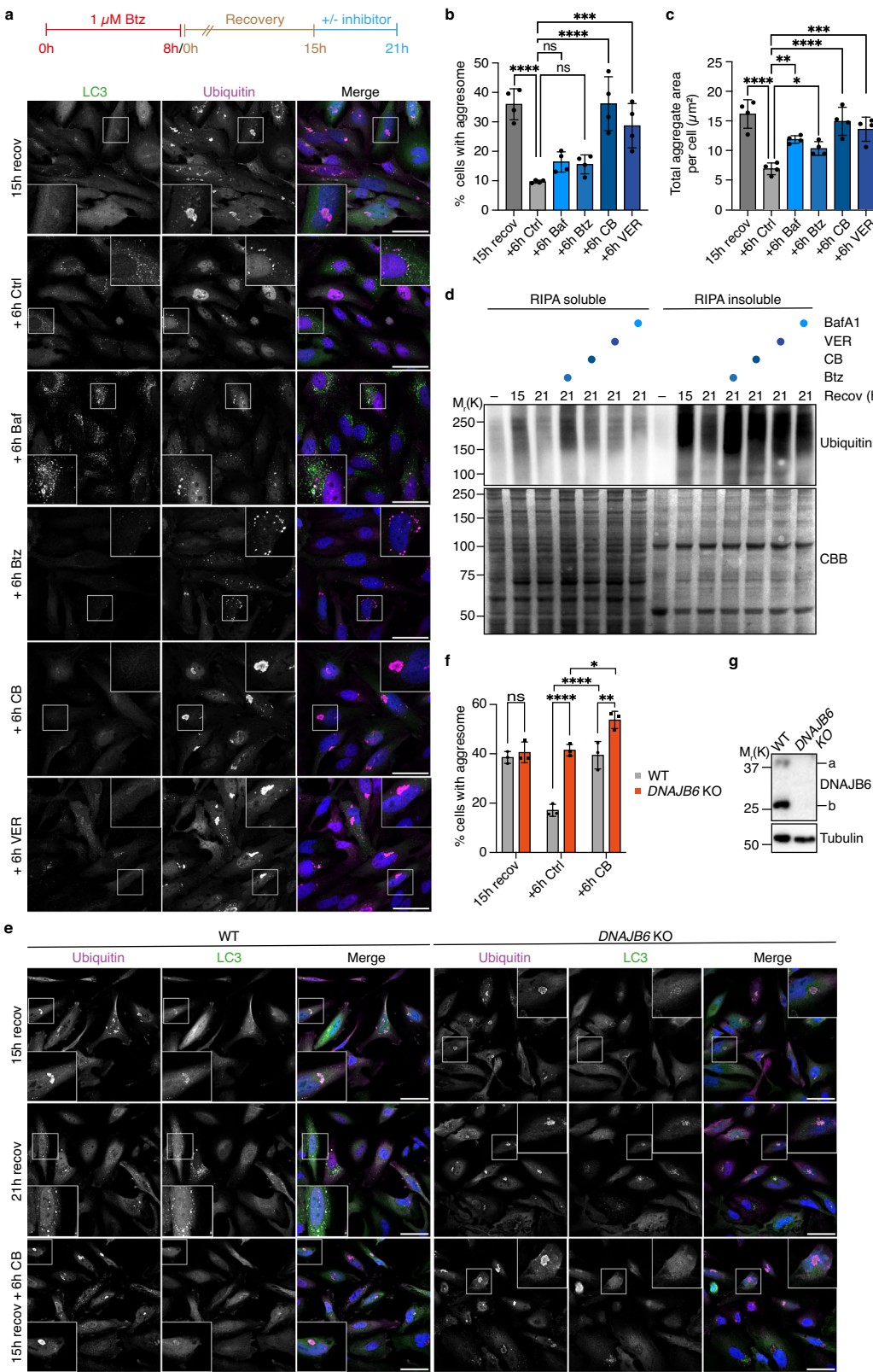

PLEK2 (pleckstrin-2), which were significantly enriched in our prox-imitome datasets (Fig. 2c, d; Supplementary Figs. 3f, 6a–c), localized to aggresomes (Supplementary Fig. 3g), and were strongly ubiquitylated around the peak of aggresome formation according to an OtUBD pulldown experiment (Fig. 6a). AMOTL2 is an angiomotin family member regulating the Hippo signaling pathway[62]. Interestingly, its ternary complex with the Hippo effectors LATS1 and YAP is disrupted upon proteasome inhibition[63]. Analyzing the degradation of AMOTL2 during aggresome clearance, we found that the addition of Btz caused an accumulation of cytoplasmic AMOTL2 signals, whereas CB or Baf had no effect (Supplementary Fig. 7a). Fractionation experiments confirmed that insoluble AMOTL2 strongly accumulated upon Btz treatment during aggresome clearance, but hardly so in the presence of CB, VER, or Baf (Supplementary Fig. 7b), indicating that AMOTL2 is a

**Fig. 3 | p97 and the Hsp70 system mediate aggresome disassembly. a** HeLa cells treated with Btz (1 μM, 8 h) were fixed after 15 h recovery or recovered for six additional hours in the absence (Ctrl) or presence of 100 nM Baf, 1 μM Btz, 1 μM CB-5083, or 10 μM VER, followed by immunostaining against ubiquitin and LC3. Scale bars, 50 μm. **b** Quantification of the percentage of cells with an aggresome in **a**. Shown is the mean ± SD from n = 4 biological replicates with ≥74 cells per condition and replicate. One-way ANOVA. **c** Same as **b**, but quantification of total aggregate area. **d** HeLa cells treated as in **b** or left untreated (−) were lysed in RIPA buffer. RIPA soluble and insoluble fractions were immunoblotted for ubiquitin, and the Coomassie staining of the membrane served as a loading control (CBB). Representative

result from five independent experiments. **e** HeLa wildtype (WT) and *DNAJB6* KO cells treated with Btz (1 μM, 8 h) were fixed after 15 h recovery or recovered for six additional hours in the absence or presence of 1 μM CB-5083, followed by immunostaining against ubiquitin and LC3. Scale bars, 50 μm. **f** Quantification of the percentage of cells with an aggresome in **e**. Shown is the mean ± SD from n = 3 biological replicates with ≥100 cells per condition and replicate. Two-way ANOVA. **g** Lysates from control and *DNAJB6* KO cells were immunoblotted for DNAJB6 (isoforms a and b) to validate KO efficiency. Representative result from two independent experiments.

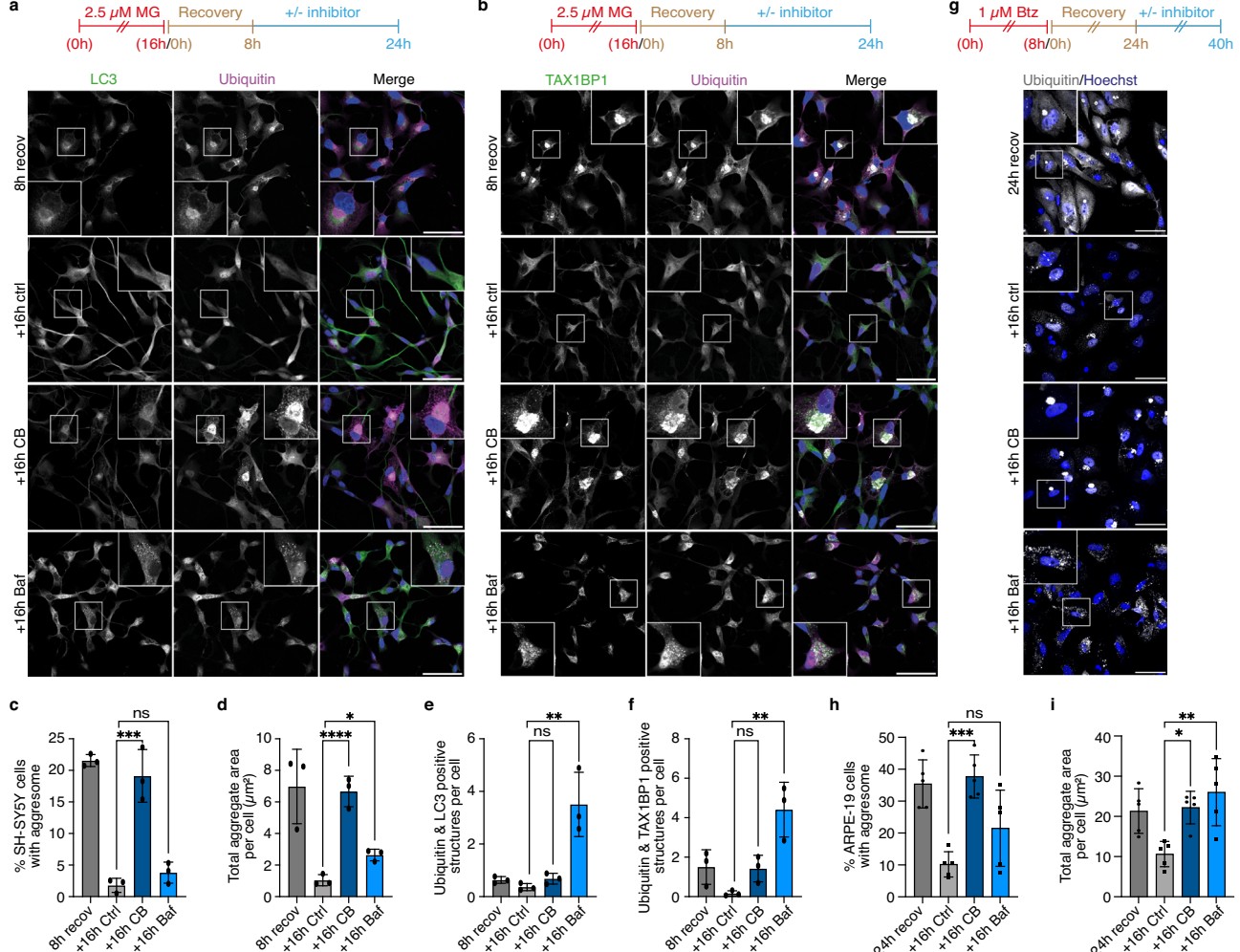

**Fig. 4 | p97 is crucial for aggresome clearance in SH-SY5Y and ARPE-19 cells. a**, **b** Retinoic acid-differentiated SH-SY5Y cells treated with MG-132 (2.5 μM, 16 h) were fixed after 8 h recovery or recovered for an additional 16 h in the absence (ctrl) or presence of 0.5 μM CB-5083 or 100 nM Baf, followed by immunostaining against ubiquitin and LC3 (**a**) or TAX1BP1 (**b**). **c** Quantification of the percentage of cells with an aggresome in **a**, **b**. Shown is the mean ± SD from n = 3 biological replicates with ≥163 cells per condition and replicate. One-way ANOVA. **d** Same as **c**, but quantification of total aggregate area. **e**, **f** Quantification of the number of ubiquitin and LC3-positive structures per cell in **a**, **e** and the number of ubiquitin and

TAX1BP1-positive structures per cell in **b**, **f**. Shown is the mean ± SD from n = 3 biological replicates with ≥81 cells per condition and replicate. One-way ANOVA. **g** ARPE-19 cells treated with Btz (1 μM, 8 h) and fixed after 24 h recovery or recovered for 16 additional hours in the absence or presence of 100 nM Baf or 0.5 μM CB-5083 prior to fixation, were immunostained against ubiquitin. Scale bars, 50 μm. **h** Quantification of the percentage of cells with an aggresome in **g**. Shown is the mean ± SD from n = 5 biological replicates with ≥100 cells per condition and replicate. One-way ANOVA. **i** Same as **h**, but quantification of total aggregate area.

mostly p97- and Hsp70-independent proteasomal substrate at aggresomes.

The putative proto-oncogene PLEK2 binds phosphatidylinositol-3-phosphates and is involved in actin remodeling processes and various signaling pathways[64,65]. The modest performance of commercially available PLEK2 antibodies in immunofluorescence applications (cf. Supplementary Fig. 3g) precluded the microscopic analysis of PLEK2

degradation during aggresome clearance. However, a RIPA-insoluble fraction of PLEK2 present at 15 h of recovery was reduced by 21 h under control conditions, but persisted in the presence of Btz, CB, VER, or Baf (Fig. 6b, c), suggesting that the clearance of insoluble PLEK2 involves the 26S proteasome, p97, Hsp70 and autophagy. To explore if PLEK2 is degraded via aggrephagy, we enriched aggrephagosomes biochemically by a differential centrifugation protocol (Supplementary Fig. 7c)

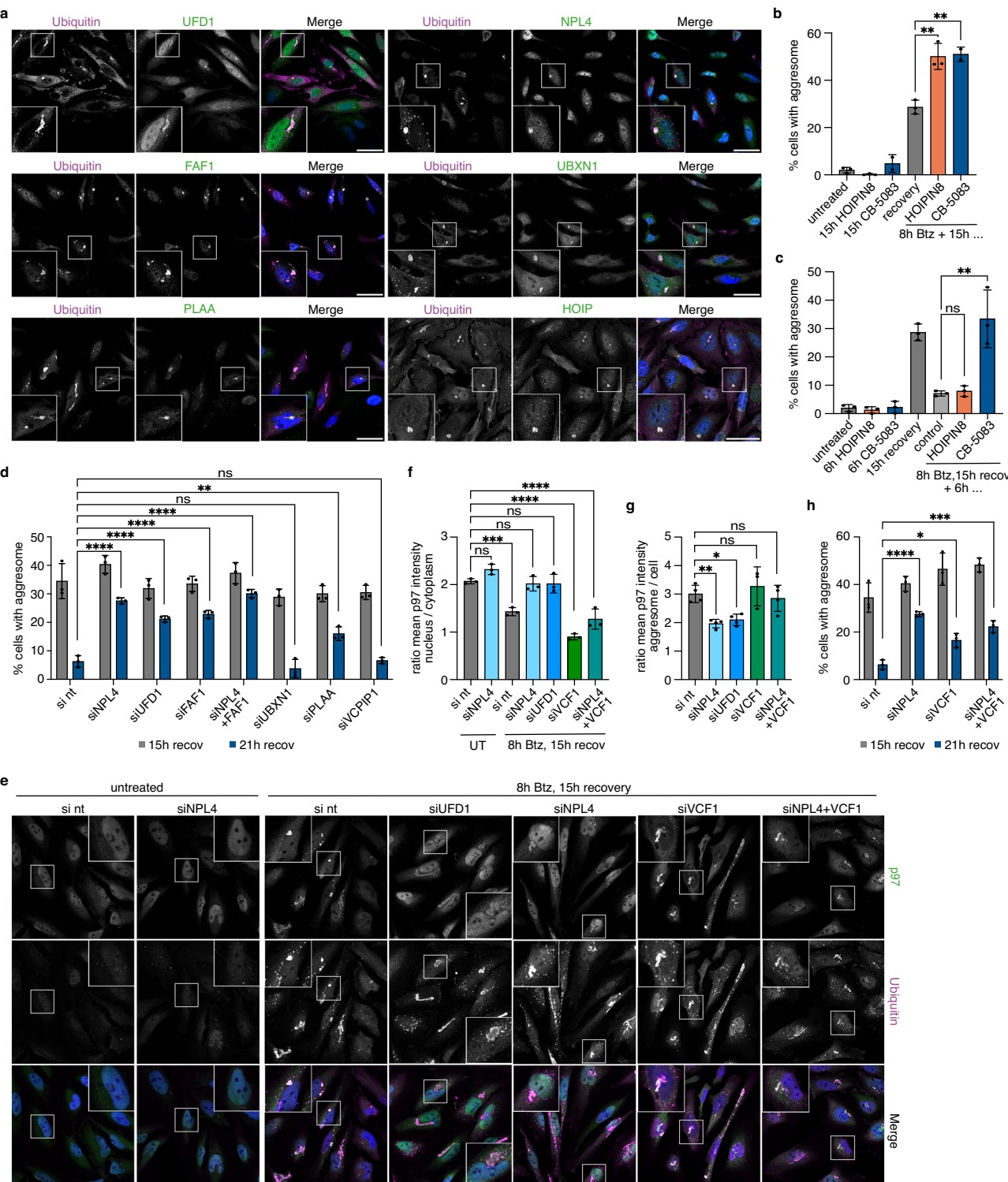

adapted from ref. 66. Fraction P1 contained the autophagosomal marker proteins TAX1BP1, p62, and lipidated LC3-II, as well as markers for lysosomes (LAMP2) and ER (Calnexin), but not mitochondria (PRDX3) or the nucleus (histone H4) (Supplementary Fig. 7d, e). Intriguingly, PLEK2 was present in the autophagosome-enriched fraction P1 under aggrephagy, but not control conditions (Fig. 6d, Supplementary Fig. 7f). Moreover, like TAX1BP1 and p62, PLEK2 in fraction P1 was resistant to Proteinase K treatment in the absence, but not presence of detergent (Fig. 6e), indicating that it was sequestered in autophagosomes and strongly suggesting that it is indeed partially degraded via aggrephagy.

A third prominent hit found in all proximitomes, FAM83D (Fig. 2c, d; Supplementary Fig. 6a–c), was reported to recruit casein kinase CK1a to the mitotic spindle and to undergo proteasomal degradation during mitotic exit[67]. Similar to PLEK2, a RIPA-insoluble fraction of FAM83D persisted in the presence of Btz, CB, or Baf during aggresome clearance (Supplementary Fig. 7g), and a fraction of FAM83D was sequestered into aggrephagosomes (Supplementary Fig. 7h–j). However, we were unable to detect its ubiquitylation during aggresome formation or clearance (Supplementary Fig. 7k), suggesting that FAM83D may not be a direct substrate of p97, but an aggregation-prone aggresomal "bystander" client

**Fig. 5 | UFD1-NPL4 and FAF1 are required for efficient aggresome clearance. a** Localization of the indicated p97 cofactors in HeLa cells recovering for 15 h from Btz treatment was analyzed by confocal immunofluorescence microscopy. Scale bars, 50 μm. Representative result from two independent experiments. **b** HeLa cells treated with Btz (1 μM, 8 h) and recovered for 15 h in the absence or presence of 30 μM HOIPIN-8 or 0.5 μM CB-5083 were immunostained for ubiquitin. Untreated cells and cells treated for 15 h with HOIPIN-8 or CB-5083 only served as controls. The percentage of cells with an aggresome was quantified; shown is the mean ± SD from n = 3 biological replicates with ≥70 cells per condition and replicate. One-way ANOVA. **c** HeLa cells treated with Btz (1 μM, 8 h) were fixed after 15 h recovery, or recovered for six additional hours in the absence or presence of 30 μM HOIPIN-8 or 1 μM CB-5083 and immunostained against ubiquitin. Untreated cells and cells treated for 6 h with HOIPIN-8 or CB-5083 only served as controls. The percentage of cells with an aggresome was quantified; shown is the mean ± SD from n = 3 biological replicates with ≥70 cells per condition and replicate. One-way ANOVA. **d** Aggresome formation at 15 h recovery from Btz treatment (gray bars) and clearance at 21 h recovery (blue bars) was quantified in control cells (si nt) and in

cells depleted of the indicated p97 cofactors. Shown in the mean ± SD from n = 3 biological replicates with ≥56 cells per condition and replicate. One-way ANOVA was performed for the 21 h recovery time point. **e** p97 recruitment to aggresomes was analyzed by confocal immunofluorescence microscopy using antibodies against p97 and ubiquitin in control cells (si nt) and cells depleted of NPL4, UFD1, VCF1, or NPL4 + VCF1 after 15 h recovery from Btz treatment. Scale bars, 50 μm. **f** Quantification of the ratios of nuclear to cytoplasmic p97 intensity in **e**. Shown is the mean ± SD from n = 3 biological replicates with ≥51 cells per condition and replicate. One-way ANOVA. **g** For the experiment shown in **e**, mean p97 intensity at aggresomes was quantified and normalized to the mean p97 intensity of the respective whole cell for the 15 h recovery condition. Shown is the mean ± SD from n = 4 biological replicates with ≥51 cells per condition and replicate. One-way ANOVA. **h** Aggresome formation at 15 h recovery from Btz treatment (gray bars) and clearance at 21 h recovery (blue bars) were quantified in control cells (si nt) and upon siRNA-mediated knockdown of NPL4, VCF1, or both. Shown is the mean ± SD from n = 3 biological replicates with ≥51 cells per condition and replicate. One-way ANOVA.

that is at least partially cleared via aggrephagy as part of aggregated protein cargo.

Insoluble cytoplasmic TDP-43 aggregates, which are hallmarks of the neurodegenerative disorders amyotrophic lateral sclerosis (ALS) and fronto-temporal dementia (FTD) as well as of multisystem proteinopathy (MSP) 1 caused by mutations in the *VCP* gene encoding p97 (refs. 68–70), were previously reported to be cleared via autophagy[19]. We therefore sought to determine the fate of a recently characterized, aggregation-prone TDP-43 variant, GFP-TDP-43 6 M&RRMm (ref. 71) (referred to here as GFP-TDP-43mut), which localized to aggresomes even when expressed at (sub-)endogenous levels (Fig. 6f, g). Whereas GFP-TDP-43mut was efficiently degraded during unperturbed aggresome clearance, it remained at persistent aggresomes upon CB treatment and accumulated in LC3- and Ub-positive aggrephagosomes and autolysosomes in the presence of Baf (Fig. 6f, h, i), indicating that GFP-TDP-43mut is a disease-relevant, p97-dependent client at aggresomes that is, at least partially, degraded via aggrephagy.

## p97-mediated disintegration of aggresomes is required for their aggrephagy

To elucidate why p97 activity is required for aggresome clearance, we quantified the formation of Ub- and LC3-positive aggrephagosomes in the absence and presence of p97 inhibitors (Fig. 7a–c). Under control conditions, most cells had cleared their perinuclear aggresome and contained around five aggrephagosomes after 21 h of recovery. In the presence of Baf, there was a strong increase in the number of aggrephagosomes and autolysosomes, but not aggresomes. By contrast, in the presence of the p97 inhibitors CB or NMS-873, aggresome break-up was significantly impaired at the expense of aggrephagosome numbers, indicating that p97 is required for normal aggrephagosome formation. Of note, the regulator of actin polymerization, profilin-2 (PFN2), which was present in our proximitomes and at disassembling aggresomes (Fig. 2c, d; Supplementary Fig. 3f, g; Supplementary Fig. 6a–c), localized to aggrephagosomes in the absence and presence of Baf, suggesting a potential role in actin-mediated aggrephagosome movement[35], but was absent from persistent aggresomes in the presence of CB (Supplementary Fig. 8a, b), thus underscoring the failure to form aggrephagosomes upon p97 inhibition.

To explore potential reasons underlying impaired aggrephagosome formation, we analyzed aggresomes persisting upon p97 inhibition for the presence of p62 and TAX1BP1, but found that both SARs localized normally to aggresomes (Fig. 7d). Moreover, the phosphorylation at serine residues 349 and 403 indicates that p62 was in an autophagy-competent state (Fig. 7e)[72,73]. Similarly, the decoration of aggresomes with K48- and K63-linked ubiquitin chains was unaffected by CB (Supplementary Fig. 8c, d).

Because inhibition of p97 was recently shown to impair autophagy initiation under starvation conditions[74], we next determined the LC3 lipidation state, but found that neither CB nor NMS-873 caused a decrease in lipidated LC3-II under our experimental conditions (Fig. 7f). Consistently, the total number of LC3 puncta representing autophagosomes was not significantly reduced in the presence of CB (Fig. 7g). Additionally, in contrast to combined Wortmannin and Baf treatment, the combined treatment with CB and Baf did not block the increase in LC3 puncta observed with Baf alone (Fig. 7g), indicating that p97 inhibition does not interfere with an early autophagosome initiation step. Together, our data suggest that global LC3 lipidation and autophagosome formation are not impaired upon inhibition of p97 for 6 h during the recovery phase. To test if p97 inhibition affected the recruitment of the phagophore formation machinery, we analyzed the localization of WIPI2 and ATG12 at aggresomes. WIPI2 is a phosphatidylinositol-3-phosphate effector downstream of the ULK1 and PI3K kinase complexes that recruits the E3-like ATG12-5-16L1 complex, catalyzing LC3 lipidation to the site of phagophore formation[75,76]. Importantly, WIPI2 and ATG12 were both present at persistent aggresomes after treatment with CB (Fig. 7h, i; Supplementary Fig. 8e). Taken together, our data indicate that the impaired aggrephagosome formation upon p97 inhibition is unlikely to be the consequence of missing factors required for selective autophagy.

Finally, we analyzed aggresome clearance in the absence and presence of CB using super-resolution structured illumination microscopy (SIM) (Fig. 7j, k; Supplementary Fig. 8f). 3D reconstructions of representative SIM images (Fig. 7j; Supplementary Movies 1–4) show that LC3 is barely localized to aggresomes around the peak of aggresome formation (15 h recovery), but is associated with aggresome fragments during unperturbed clearance to form aggrephagosomes, which strongly accumulated in the presence of Baf. By contrast, only little LC3 was present at the persistent, robust aggresomes observed after 15 h of recovery, followed by 6 h of p97 inhibition, indicating that the p97-mediated disintegration of aggresomes is a prerequisite for the subsequent sequestration of aggresomal fragments into aggrephagosomes.

## Discussion

In this study, we characterized the clearance of endogenous aggresomes induced by proteasomal inhibition. The SAR TAX1BP1 had been previously shown to mediate the aggrephagy of peripheral and polyQ-containing aggregates[41]. Here we show, to our knowledge for the first time, that TAX1BP1 is also indispensable for the clearance of mature, endogenous aggresomes, underscoring its central role in the removal of various types of aggregates. The TAX1BP1 proximitome presented here identified PQC factors, autophagy machinery, and client proteins associated with endogenous aggresomes and aggresome-derived

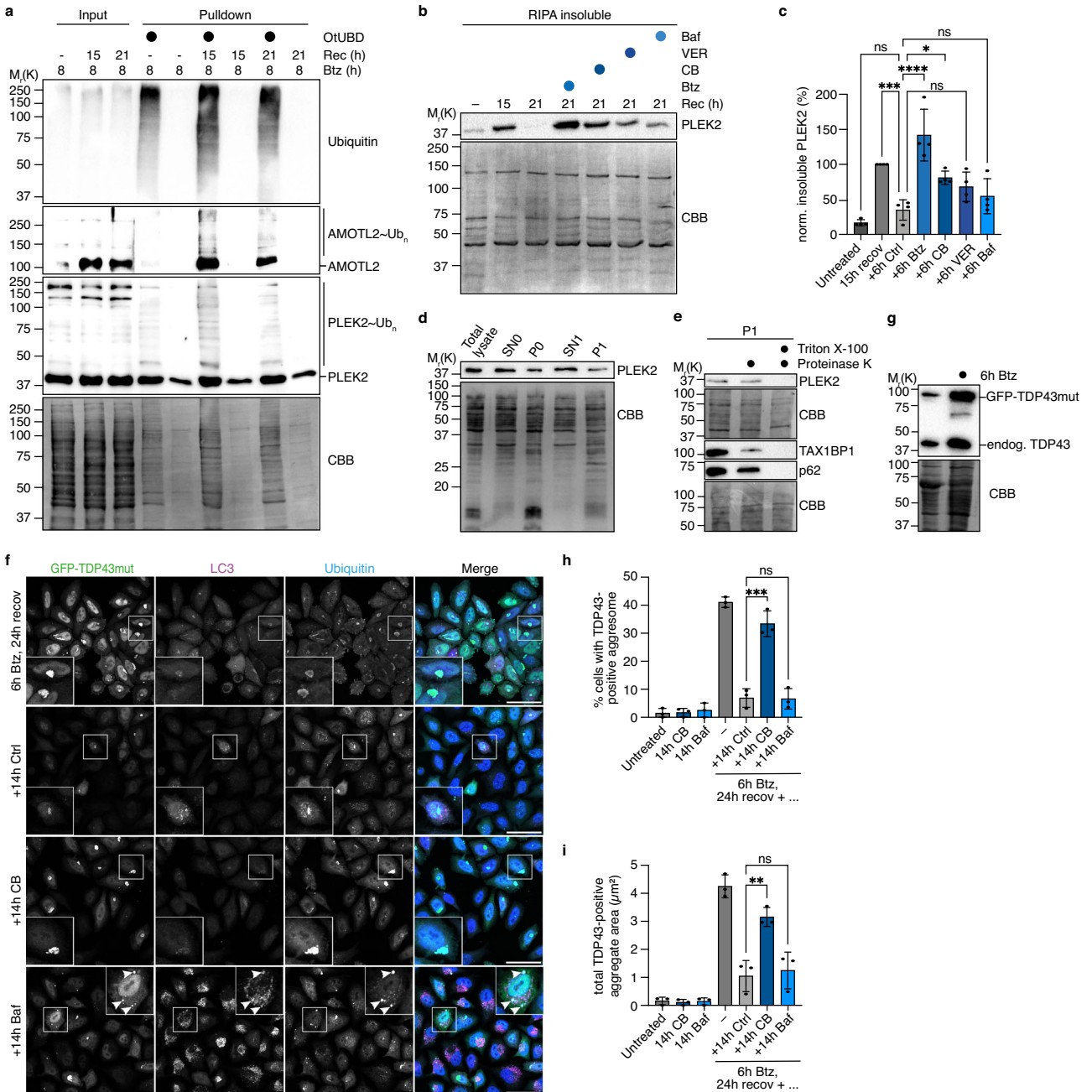

**Fig. 6 | Aggresomal proteins are degraded via different pathways. a** HeLa cells treated with Btz (1 µM, 8 h) and recovered for 0 h, 15 h, and 21 h were lysed and subjected to a pulldown of ubiquitylated proteins using OtUBD beads. Input and pulldown samples were immunoblotted for ubiquitin, AMOTL2, and PLEK2; the Coomassie-stained membrane served as a loading control (CBB). Representative result from two independent experiments. **b** HeLa cells treated with Btz (1 µM, 8 h) and harvested after 15 h recovery, or recovered for six additional hours in the absence or presence of 1 µM Btz, 1 µM CB-5083, 10 µM VER, or 100 nM Baf were lysed in RIPA buffer. Untreated cells served as negative controls. RIPA-insoluble fractions were immunoblotted for PLEK2; the Coomassie-stained membrane served as a loading control (CBB). Shown is a representative western blot of four independent biological replicates. **c** Quantification of insoluble PLEK2 levels in **b**. Bars represent the mean fold change of insoluble PLEK2 relative to the levels at 15 h recovery ± SD (n = 4 biological replicates). One-way ANOVA. **d** HeLa cells treated with Btz (1 µM, 8 h) were recovered for 15 h and incubated with 100 nM Baf for six additional hours prior to cell lysis and isolation of an autophagosome-enriched fraction. The fractions of the isolation protocol depicted in Supplementary Fig. 7c

were immunoblotted for PLEK2; the Coomassie-stained membrane served as a loading control (CBB). Representative result from two independent experiments. **e** Immunoblot analysis of autophagosome-enriched fraction (P1) after Proteinase K treatment with the indicated antibodies. Coomassie-stained membranes served as loading controls (CBB). Representative result from three independent experiments. **f** HeLa Flp-In TRex cells expressing GFP-TDP-43mut were treated with Btz (1 µM, 6 h) and fixed after 24 h recovery, or recovered for 14 additional hours in the absence or presence of 0.5 µM CB-5083 or 100 nM Baf, followed by immunostaining against ubiquitin and LC3. Scale bars, 50 µm. In the bottom row, arrowheads highlight GFP-, LC3-, and ubiquitin-positive structures. **g** Expression of endogenous TDP-43 and GFP-TDP-43mut in HeLa Flp-In TRex cells that were left untreated or were treated for 6 h with Btz was analyzed by immunoblot. Representative result from two independent experiments. **h** Quantification of the percentage of cells with a TDP-43-positive aggresome in **f**. Shown is the mean ± SD from n = 3 biological replicates with ≥150 cells per condition and replicate. One-way ANOVA. **i** Same as **h**, but quantification of total TDP-43-positive aggregate area.

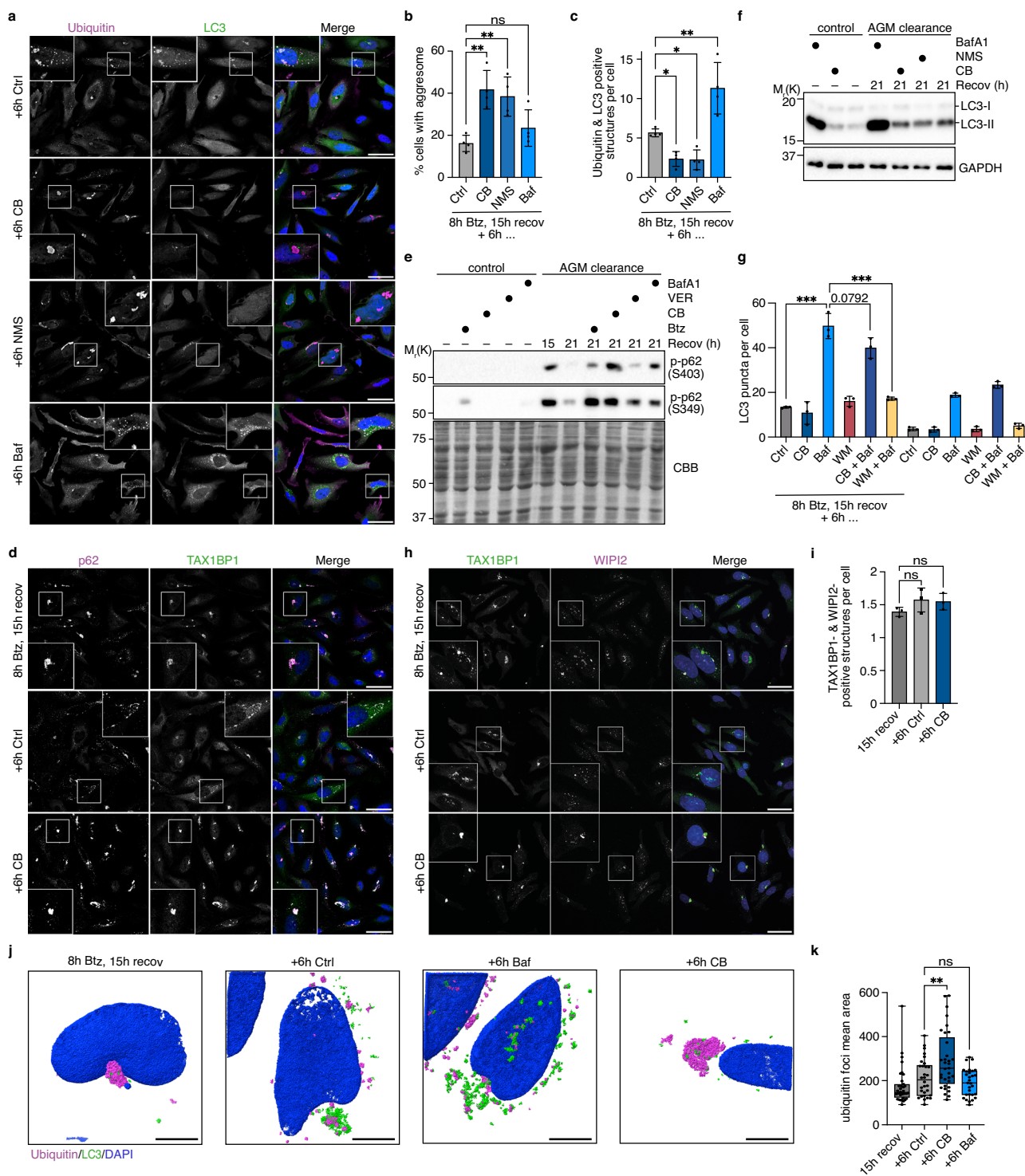

aggrephagosomes. By contrast, a recent APEX2-TAX1BP1 proximitome study profiling basal autophagosome content as well as Btz-induced cargo candidates under conditions that did not induce significant aggresome formation identified several SARs and other autophagy factors, but only few of the chaperones found in our datasets and neither the 26S proteasome nor p97 (ref. 77). The limited overlap with our TAX1BP1 proximitome is probably due to the different time points of analysis and experimental conditions used in that study.

Our experimental approach differs from previous studies in three important aspects that, combined, enabled us to specifically address aggresome clearance. First, we refrained from inducing aggresomes by ectopic overexpression of aggregation-prone proteins, which results

in non-uniform, artificially high expression levels and is difficult to precisely shut off. Instead, we applied a defined period of proteasomal inhibition by Btz. Second, we carefully determined the peak of aggresome formation, which, unexpectedly, was reached only 15 h after Btz washout, defining an explicit starting point of aggresome clearance. Third, wherever possible, we used established pharmacological inhibitors of PQC factors to analyze aggresome clearance in the absence of confounding effects of genetic de(p)letion.

Using this strategy, we uncovered a clear hierarchy of PQC systems during aggresome clearance (Fig. 8): p97 and Hsp70 are required first for the disintegration of the aggresome into smaller fragments, presumably by disaggregation of aggresomal clients such as PLEK2

**Fig. 7 | The p97-mediated disintegration of aggresomes is required for their packing into autophagosomes. a** HeLa cells treated with Btz (1 μM, 8 h) were fixed after 15 h recovery, or recovered for six additional hours in the absence or presence of 1 μM CB-5083, 5 μM NMS-873, or 100 nM Baf prior to fixation and analysis by confocal microscopy using antibodies against ubiquitin and LC3. Scale bars, 50 μm. **b** Quantification of the percentage of cells with an aggresome in **a**. Shown is the mean ± SD from n = 4 biological replicates with ≥60 cells per condition and replicate. One-way ANOVA. **c** Quantification of the number of ubiquitin- and LC3-positive structures per cell in **a**. Shown is the mean ± SD from n = 4 biological replicates with ≥60 cells per condition and replicate. One-way ANOVA, corrected for multiple comparisons by Holm–Šidák's multiple comparison test. **d** HeLa cells treated with Btz (1 μM, 8 h) were fixed after 15 h recovery, or recovered for six additional hours in the absence or presence of 1 μM CB-5083 prior to fixation and analysis by confocal microscopy using antibodies against p62 and TAX1BP1. Scale bars, 50 μm. Representative result from two independent experiments. **e** HeLa cells treated with Btz (1 μM, 8 h) were lysed after 15 h recovery, or recovered for six additional hours in the absence or presence of 1 μM Btz, 1 μM CB-5083, 10 μM VER, or 100 nM Baf prior to lysis in SDS sample buffer and immunoblotting for p-p62 S403 and S349. Control samples were taken without prior Btz treatment. AGM, aggresome. Representative result from three independent experiments. **f** HeLa cells treated with Btz (1 μM, 8 h) and recovered for 21 h in the absence or presence of 100 nM Baf, 1 μM CB-5083, or 5 μM NMS-873 for the last 6 h were immunoblotted for LC3 and GAPDH. Control samples were taken without prior Btz treatment. AGM, aggresome. Representative result from two independent experiments. **g** HeLa cells

treated with Btz (1 μM, 8 h) and recovered for 21 h in the absence or presence of 1 μM CB-5083, 100 nM Baf, 2 μM Wortmannin, or combinations of these inhibitors in the last 6 h were immunostained against LC3. Control samples without prior Btz treatment were included. The number of LC3 puncta per cell was quantified. Shown in the mean ± SD from n = 3 biological replicates with ≥60 cells per condition and replicate. Unpaired, two-tailed Student's t-tests. **h** HeLa cells treated with Btz (1 μM, 8 h) and fixed after 15 h recovery, or recovered for six additional hours in the absence or presence of 1 μM CB-5083 were analyzed by confocal microscopy using antibodies against WIPI2 and TAX1BP1. Scale bars, 50 μm. **i** Quantification of the number of WIPI2 and TAX1BP1 double-positive structures per cell in **h**. Shown is the mean ± SD from n = 3 replicates with ≥55 cells per condition and replicate. One-way ANOVA. **j** HeLa cells treated with Btz (1 μM, 8 h) and fixed after 15 h recovery, or recovered for six additional hours in the absence or presence of 1 μM CB-5083 or 100 nM Baf were analyzed by 3D-SIM using antibodies against ubiquitin and LC3. Shown are representative 3D renderings of image stacks. Scale bars, 10 μm. **k** Box-and-whisker plot showing the mean area of the ubiquitin foci from a total of 29–36 cells per condition, of which 12 representative cells per condition are shown in Supplementary Fig. 8f. The box shows the 25th and 75th percentiles and the median, and the whiskers reach from the minimum to the maximum value; each individual value is plotted as a point. One-way ANOVA was performed over three independent experiments (n = 30 cells for +6 h Ctrl, n = 33 cells for +6 h CB, n = 29 cells for + 6 h Baf), corrected for multiple comparisons by Dunnett's multiple comparison test.

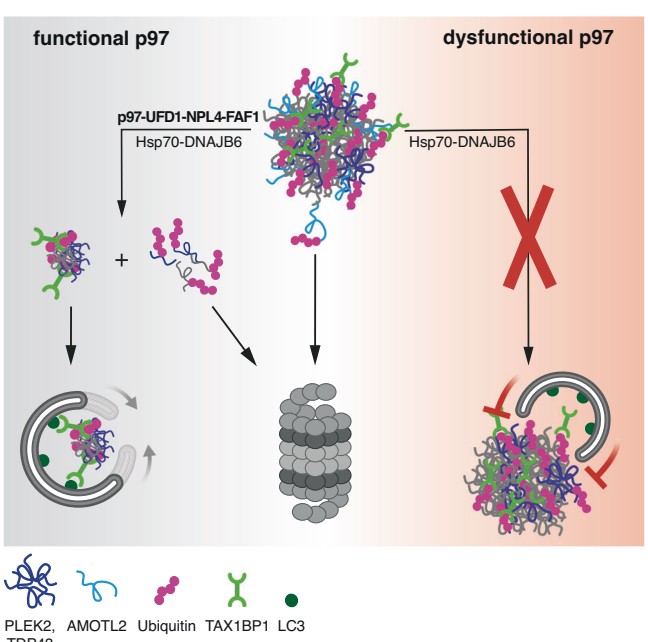

**Fig. 8 | Model of p97-dependent, piecemeal autophagy of aggresomes.** See the "Discussion" section for details. Figure partially created in BioRender. Müller, P. (2025) [https://BioRender.com/fj6r4sf].

and TDP-43. Aggresome fragments are subsequently cleared by TAX1BP1-dependent aggrephagy. Even though the 26S proteasome on its own can degrade easily accessible aggresomal proteins such as AMOTL2, as well as substrates solubilized by p97 and/or Hsp70, it cannot mediate aggresome clearance in the absence of either of these disaggregases. While we presently cannot exclude the possibility that p97 and Hsp70 must remove (yet unknown) specific, critical proteins to initiate aggresome disintegration, we favor the hypothesis that the disaggregases target client proteins stochastically and that aggresomes break up as a mere consequence of ongoing protein removal. It remains to be determined if p97 and Hsp70 act in this process cooperatively on the same substrate proteins, as has been suggested in the

case of Tau fibril disaggregation[24] and shown for p97-related Hsp100 family disaggregases[78,79], or if they act in parallel on distinct substrates.

Interestingly, our proximitome data identified DNAJB6 as the most strongly enriched JDP. Previously, a combination of class A and B JDPs, specifically DNAJA2 and DNAJB1, was shown to endow Hsp70 chaperones with the ability to efficiently disassemble fibrillar aggregates[80,81]. While the TAX1BP1 proximitome also contained DNAJB1 (and DNAJA1), depletion of DNAJB6 caused a complete clearance defect, indicating that the critical function of DNAJB6 in aggresome disaggregation cannot be taken on by DNAJB1 and, more generally, that distinct class B JDPs possess preferences for specific types of aggregates[45,82]. Of note, mutations in *DNAJB6* cause limb-girdle muscular dystrophy type 1D (LGMD1D) and potentially FTD (reviewed in ref. [83]), and LGMD1D muscle fibers exhibit rimmed vacuoles and TDP-43- and ubiquitin-positive inclusions resembling those found in MSP1 (refs. [70,84]). Considering the requirement for p97 and DNAJB6 during aggresome clearance reported here, it is tempting to speculate that failure of mutant p97 and DNAJB6 to efficiently clear aggresomes contributes to the pathogenesis of LGMD1D and MSP1.

A recent study showing that p97 is required for aggresome formation did not report a statistically significant aggresome clearance defect upon inhibition of p97 (ref. [37]). However, in these experiments, CB was added right after the Btz treatment, suggesting that it also affected the ongoing formation of aggresomes. It is likely that this early addition of CB obscured the virtually complete block in clearance seen under our conditions, where CB was added after completion of aggresome formation.

Our study demonstrates, to the best of our knowledge for the first time, that aggresome clearance requires the p97 cofactors UFD-NPL4 and FAF1. These cofactors are not simply needed to recruit p97 to aggresomes, as the restored aggresome localization of p97 upon co-depletion of NPL4 and VCF1 was insufficient to effectuate aggresome clearance (Fig. 5e–h). Rather, UFD1-NPL4 and FAF1 are likely to functionally cooperate in the canonical priming of ubiquitylated proteins at aggresomes for p97-mediated unfolding/disaggregation[85]. Such cooperation of UFD1-NPL4 and FAF1 is reminiscent of their reported nuclear functions in chromatin-associated degradation and CMG helicase disassembly[59,60,86] and lends further support to the proposal that the cooperative assembly of p97 complexes containing UFD1-NPL4 and FAF1 (or related cofactors with a UBA-UBX domain architecture) is a general feature of p97 targeting ubiquitylated substrates[60].

p97 and its cofactor HOIP, the catalytic subunit of the LUBAC E3 ligase catalyzing the formation of M1-linked Ub chains[50], were recently shown to be recruited in a feed-forward loop to various disease-associated protein aggregates in order to promote their proteasomal degradation[26]. Our data demonstrate that HOIP activity is needed in the early proteotoxic stress response to limit the formation of aggresomes, but not during aggresome clearance. Moreover, HOIP was not required for the recruitment of p97 to aggresomes, strongly suggesting that HOIP and M1 chains are involved in the clearance of early, peripheral aggregates, whereas UFD1-NPL4 and FAF1 are critical for the disintegration of mature aggresomes. While the aggresome localization and proximitomes of UBXN1 and PLAA implicate these cofactors in aggresome turnover as well (Fig. 5a, Supplementary Fig. 6a–c)[37], their exact roles remain to be uncovered.

Recent studies highlighted the importance of cargo liquidity for autophagosome formation[87,88]. In 'fluidophagy', protein-free liquid droplets are sequestered into autophagosomes either in a piecemeal manner or in toto by virtue of the intrinsic curvature of the phagophore membrane[88], whereas the selective autophagy of semi-liquid protein condensates requires SARs 'floating' on their surface for efficient sequestration[87]. Accordingly, the reduced liquidity[89] and large size of aggresomes are predicted to present an obstacle to their piecemeal and/or SAR-mediated incorporation into autophagosomes[87]. Our finding that the Hsp70- and p97-dependent disintegration of aggresomes is a prerequisite for the sequestration of the resulting fragments into aggrephagosomes is in full agreement with this hypothesis.

## Methods

All materials used, including antibodies, cell lines, and reagents, are listed in the Supplementary Information file (Supplementary Table 1).

### Growth, transfection, and treatment of mammalian cells

ARPE-19 and SH-SY5Y cells were maintained in Dulbecco's modified Eagle's medium (DMEM)/F-12 media supplemented with 10% fetal bovine serum and 1% penicillin/streptomycin. All other cell lines were maintained in DMEM supplemented with 10% fetal bovine serum and 1% penicillin/streptomycin. All cell lines were cultured in a humidified atmosphere with 5% $CO_2$ at 37 °C. 1.5 µg/ml puromycin was added to the culture media for knockout cell pools. Mycoplasma contamination was regularly checked by performing PCR-based tests.

For differentiation of SH-SY5Y cells into a neuron-like phenotype, approx. 25,000 to 50,000 cells/ml were seeded in 12-well plates and grown on coverslips precoated with 50 µg/ml Poly-D-Lysine (Thermo Fisher). 24 h to 48 h after seeding, serum-containing medium was replaced with Neurobasal medium (containing B27 supplement and GlutaMAX) and 10 µM all-*trans*-retinoic acid (ATRA) to induce differentiation for 3 days, refreshing the medium every 16–24 h. The progression of differentiation was monitored microscopically by examining the morphological development of neurite outgrowth.

For siRNA-mediated depletion of the different p97 cofactors, cells were seeded in 12-well plates and transfected with the respective siRNA (50 nM final concentration) using 1.2 µl Oligofectamine diluted in 100 µl Opti-MEM following the manufacturer's protocol. After 20 h, the medium was changed, and the knockdown was continued for a total of 72 h.

As indicated in the respective figure legends, cells were treated for 6–14 h with the indicated concentrations of Bortezomib, CB-5083, NMS-873, VER-155008, Bafilomycin A1, Wortmannin, HOIPIN-8, Tubacin, or Tubastatin A.

### Generation of HeLa knockout cell pools

To generate *TAX1BP1* and *DNAJB6* knockout cell pools, gRNAs targeting the human *TAX1BP1* or *DNAJB6* genes were designed using CHOPCHOP (ref. 90), ordered as complementary oligos with BpiI-compatible overhangs (Supplementary Table 1), and ligated with BpiI-digested pLentiCRISPRv2 (Addgene #52961, gift from Feng Zhang)[91]. Non-human-target and *HOIP* targeting gRNAs were designed and provided in pLentiCRISPRv2 by Manuel Kaulich. When more than one gRNA was selected per target, the corresponding pLentiCRISPRv2 constructs were pooled before lentivirus production. After production of recombinant lentiviruses in HEK293T cells by co-transfection of the pLentiCRISPRv2 constructs with pMD2.G and psPAX2 (Addgene #12259 and #12260; gifts from Didier Trono), HeLa cells at 50%–70% confluence were transduced with lentivirus-containing supernatant (filtered through a 0.45-µm filter) mixed with polybrene (8 µg/ml final concentration). 48 h after transduction, cells were split in culture medium containing 1.5 µg/ml puromycin, and the cell pools were kept under constant selection.

### Generation of stable HeLa Flp-In-TRex cell lines

HeLa Flp-In-TRex cells were seeded in 6-well plates and co-transfected with 1.2 µg pOG44 and 0.6 µg pcDNA5 6M&RRMm TDP-43 (ref. 71) at 80% cell confluence using Lipofectamine 2000 according to the manufacturer's instructions. 48 h post-transfection, the growth medium was changed to selection medium containing 5 µg/ml puromycin and 400 µg/ml hygromycin. Cell pools were recovered for around two weeks and kept under constant puromycin and hygromycin selection. 6M&RRMm TDP-43 expression was validated by immunoblotting after induction with 1 µg/ml tetracycline for 24 h. 6M&RRMm TDP-43 expression was induced with 1 ug/ml tetracycline for 24 h before starting proteasome inhibitor treatment, and tetracycline was omitted from the media once mature aggresomes had formed after 24 h of recovery.

### HeLa cells expressing free TurboID, TurboID-UBXN1, and PLAA-TurboID

Free TurboID was amplified from V5-TurboID-NES/pcDNA3 (Addgene #107169; gift from Alice Ting)[43] and assembled into AgeI/MluI digested pRRLSin.cPPT.PGK. Similarly, TurboID together with UBXN1 (amplified from pAB275) and PLAA (amplified from Addgene #85669) were assembled into double-digested pRRLSin.cPPT.PGK using Gibson Assembly (New England Biolabs) according to the manufacturer´s instructions. All primers used during cloning are listed in Supplementary Table 1. Correct amplification was confirmed using Sanger sequencing.

After production of recombinant lentiviruses in HEK293T cells by co-transfection of the cloned pRRLSin.cPPT.PGK constructs with pMD2.G and psPAX2, HeLa cells at 50%–70% confluence, were transduced with lentivirus-containing supernatant mixed with polybrene (8 µg/ml final concentration). 48 h after transduction, cells were split in culture medium containing 1.5 µg/ml puromycin, and the cell pools were kept under constant selection.

### Generation of a TurboID-TAX1BP1 knock-in cell line

Using CHOPCHOP, a gRNA targeting the proximity of the N terminus of TAX1BP1 was selected (5′- CTGCAATGGGACTTCTTGAA - 3′) and ordered as complementary oligos with BpiI-compatible overhangs, followed by ligation into BpiI-digested pSpCas9(BB)-2A-Puro[92]. 650 bp long homology arms on both sides of the edit site were designed, consisting of the chromosomal sequence around the N terminus with silent mutations making the repair template resistant against the sgRNA, and ordered as gblocks from IDT DNA (Supplementary Table 1). V5-TurboID was amplified from V5-TurboID-NES/pcDNA3 (Addgene #107169) with primers listed in Supplementary Table 1 and ligated together with both gblocks into the EcoRI/HindIII-digested pUC19 vector via Gibson assembly. HeLa cells were seeded in a 6-well plate and co-transfected upon reaching 80% cell confluence with 1 µg TAX1BP1 sgRNA in pSpCas9-Puro and 1 µg of TAX1BP1 repair template in pUC19 using PEI in a 1:3 DNA:PEI ratio. 24 h post-transfection, the

medium was changed, and 48 h post-transfection, cells were split in selection medium containing 1.5 µg/ml puromycin. After around 30 h, the selection medium was removed, and cells were recovered for four days before single-cell dilution and seeding single clones in 96-well plates. Following growth and expansion for three weeks, clones were screened by immunoblotting, and correct insertion of the repair template was confirmed by extraction of genomic DNA and Sanger sequencing of a PCR amplicon spanning the Cas9 cleavage site, using the primers listed in Supplementary Table 1. Both homo- and heterozygous knock-in clones were isolated and compared in biotinylation efficiency and cell growth, and a heterozygous knock-in clone was chosen for the experiments shown.

A homozygous TAX1BP1 knockout clone that had not repaired the Cas9-generated indel was used for the rescue experiment shown in Supplementary Fig. 2d–g (referred to as "R-TAX1BP1"). To this end, lentivirus was produced and transduced as described above using psPAX2, pMD2.G, and pINDUCER TAX1BP1-mScarlet. After 48 h, the medium was exchanged, and cells were selected with 1000 µg/ml neomycin G418 for 72 h. After selection, cells were kept in 400 µg/ml neomycin G418. To induce the ectopic expression of TAX1BP1-mScarlet, cells were treated with 50 ng/ml doxycycline for 16 h, and expression was validated by western blot and immunofluorescence analysis.

## Proteasome activity measurement

Cells were seeded in 96-well plates (approx. 7,500 HeLa cells per well), treated as indicated, and assayed using the Proteasome-Glo™ Chymotrypsin-Like Cell-Based Assay (Promega) according to the manufacturer's protocol. Triplicate samples were run to measure luciferase activity using a plate reader (Tecan Infinite 200). After measuring luminescence, cells were washed with PBS, fixed using 3.7% formaldehyde in PBS for 15 min at RT and washed twice with PBS, permeabilized with 0.2% Triton X-100 in PBS for 10 min at RT, washed with PBS, and incubated with 2.5 µg/ml Hoechst 33342 in PBS for 10 min at RT in the dark. After washing the cells two times with PBS, images were taken with an Operetta CLS High-Content Imaging System (Revvity) with 20-fold magnification and analyzed using the Harmony High-Content and Imaging Analysis Software (Revvity). The proteasome activity was derived from the absolute luminescence per cell and normalized to the untreated control sample.

## Immunoblotting

Protein samples were separated by SDS-PAGE and transferred onto PVDF membrane (Millipore) by semi-dry blotting using 1× Tris-glycine buffer (192 mM glycine, 25 mM Tris base, pH 8.3) supplemented with 20% methanol. The membrane was then stained for 1 min in Coomassie staining solution (40% ethanol, 10% acetic acid, 0.1% Coomassie Blue R-250), destained for 10 min in destaining solution (40% ethanol, 10% acetic acid), imaged, and completely destained for 5 min in 100% methanol. The membrane was blocked with 5% milk in TBST (50 mM Tris-HCl pH 7.5, 150 mM NaCl, 0.1% Tween 20) and incubated with the indicated primary antibody in blocking solution or 3% BSA overnight at 4 °C. The membrane was washed with TBST (3 × 10 min), incubated with HRP-conjugated secondary antibody (Dianova) diluted 1:7,500 in blocking solution for 1 h at room temperature (RT), washed again with TBST (3 × 10 min), and incubated with Clarity Western ECL Substrate (Bio-Rad). Chemiluminescence signals were detected using the Gel Doc XR+ System (Bio-Rad), and immunoblot images were processed and analyzed with Image Lab (Bio-Rad).

## RIPA fractionation

Cells were seeded in 6-well plates (180,000 HeLa cells per well), treated as indicated, washed two times with cold PBS, and harvested by adding 90 µl RIPA buffer containing protease inhibitors into the well. Lysates were scratched from the well, transferred to a reaction tube, and snap frozen in liquid nitrogen. After thawing and incubation on ice for 20 min, lysates were centrifuged for 20 min at $21,000 \times g$ at 4 °C. The soluble lysate was transferred to a fresh tube, used for protein concentration determination, and for preparation of an SDS sample (soluble fraction) by adding a five-fold concentrated SDS sample buffer (250 mM Tris-HCl pH 6.8, 10% [w/v] SDS, 30% glycerol, 500 mM dithiothreitol, bromophenolblue). The pellet was washed with 90 µl RIPA lysis buffer, vortexed, and centrifuged again for 15 min at 4 °C and $21,000 \times g$. Washing buffer was aspirated, and the pellet was resuspended in 30 µl two-fold concentrated SDS sample buffer and heated, together with the sample from the soluble fraction, at 95 °C for 5–10 min, followed by SDS-PAGE and immunoblot analysis. Loading volumes were adjusted to determine protein concentrations.

To quantify RIPA-insoluble PLEK2 levels, individual PLEK2 bands and their corresponding CBB lanes were manually identified using Image Lab. The intensity of the bands was measured, and the intensity values for the PLEK2 bands were normalized to those of the CBB lanes.

## OtUBD pulldown experiments

OtUBD pulldowns of ubiquitylated proteins under denaturing conditions were performed essentially as described[93]. Briefly, Cys-His6-OtUBD encoded by pET21a-cys-His6-OtUBD (Addgene #190091; gift from Mark Hochstrasser) was expressed in *E. coli* and purified via Ni-NTA affinity chromatography, followed by size exclusion chromatography (HiLoad 26/600 Superdex 75; Cytiva) in 50 mM Tris•HCl pH 7.5, 150 mM NaCl, 5% glycerol, 1 mM TCEP. Covalently linked OtUBD resin was prepared by conjugating Cys-His6-OtUBD to SulfoLink coupling resin (Thermo Fisher Scientific) according to the manufacturer's protocol. SulfoLink coupling resin incubated with 50 mM L-cysteine was used as a negative control.

For OtUBD pulldown experiments of ubiquitylated proteins under denaturing conditions, cells were grown to 80-90% confluence, washed twice with cold PBS, scraped in urea-containing lysis buffer (8 M Urea, 50 mM Tris-HCl pH 7.4, 100 mM NaCl, 1 mM EDTA; freshly supplemented with protease inhibitor cocktail tablets (Roche), 20 mM NEM (in EtOH), 2 mM TCEP, 1 mM PMSF, 50 µM PR-619) and incubated on ice for 30 min, followed by sonication for 10 min using a SONOREX ultrasonic bath (Bandelin). After centrifugation (15 min, $18,000 \times g$, 4 °C), cleared lysates were diluted 1:1 with lysis buffer lacking urea to adjust a final concentration of 4 M urea. After determination of protein concentration (Pierce BCA Protein Assay Kit), 5%–10% of the cleared lysate was taken as an input sample and supplemented with five-fold concentrated SDS-PAGE sample buffer, followed by heat denaturation at 95 °C. For the pulldown, lysate corresponding to 1.6 mg protein and 50 µl (bed volume) of OtUBD or negative control resin was used. The resins were equilibrated by three washes with 500 µl of OtUBD column buffer (50 mM Tris-HCl pH 7.5, 150 mM NaCl, 1 mM EDTA, 0.5% Triton X-100, 10% glycerol). The lysate was added to the equilibrated resins and incubated at 4 °C for 2.5 h with end-over-end rotation. After incubation, the resin was centrifuged (3 min, $1000 \times g$, 4 °C), and the unbound fraction was discarded. The resin was resuspended in 300 µl of OtUBD column buffer and washed sequentially with 750 µl of Wash buffer 1 (50 mM Tris-HCl, 150 mM NaCl, 0.05% Tween 20, pH 7.5) and Wash buffer 2 (50 mM Tris-HCl, 1 M NaCl, pH 7.5). After a final wash with 100 µl of pure water, the proteins were eluted by shaking at 65 °C for 30 min in 2x SDS-PAGE sample buffer.

## Isolation of an autophagosome-enriched fraction

Cells were grown to 80% confluence. All subsequent steps were performed on ice or at 4 °C with pre-cooled solutions. The cells were washed twice with 1× PBS, scraped gently from the plate in 10 mL 1× PBS, and transferred into a 15 mL Falcon tube. Afterwards, the cells were centrifuged at $500 \times g$ for 3 min and the supernatant was discarded. The cell pellet was washed once with 1× PBS, centrifuged again

at 500 × *g* for 3 min, and the supernatant was discarded again. Subsequently, the cells were resuspended in 1 mL 1× PBS supplemented with protease inhibitor cocktail tablets (Roche) and dounced in a cell homogenizer (Isobiotec) with an 18 micron ball with 20 strokes. The total lysate was centrifuged at 3000 × *g* for 10 min. The supernatant (SN0) was transferred into a fresh tube. The pellet (P0) was resuspended in 500 μL 1× PBS supplemented with protease inhibitor cocktail tablets (Roche). Following this, the supernatant (SN0) was centrifuged at 150,000 × *g* (Beckman Coulter Optima MAX-XP; Rotor MLA-130) for 30 min. The resulting supernatant (SN1) was discarded, and the resulting pellet (P1) was resuspended in 200 μL 1× PBS. From all the different isolation steps, Western Blot samples were taken and subsequently analyzed by immunoblotting.

### Proteinase K digestion

All steps were performed on ice. The autophagosome-enriched fraction (P1) was incubated with 5 μg/mL proteinase K for 15 min. As a positive control, vesicles were ruptured with 1% Triton X-100 for 15 min prior to incubation with proteinase K. As a negative control, the autophagosome-enriched fraction (P1) was incubated with 5 μg/mL BSA for 15 min instead of proteinase K. Proteinase K was inactivated by the addition of 1 mM PMSF. Samples were directly taken up in SDS sample buffer, boiled at 95 °C for 5 min, and subsequently analyzed by immunoblotting.

### Enrichment of biotinylated proteins

After adding 50 μM biotin onto the cells for 20–60 min, cells were washed five times on ice using 10 ml ice-cold PBS per 10 cm dish. On plate-lysis was performed by adding 300 μl RIPA Lysis Buffer (Thermo Fisher Scientific) supplemented with a protease inhibitor cocktail tablet and 1 mM PMSF per 10 cm dish. Lysates were incubated for 20 min on ice, sonicated six times for 9 s in a Branson Sonifier using a microtip at 35% output control, and cleared by centrifugation (21,000 × *g*, 20 min, 4 °C). After determination of the protein concentration using Pierce BCA Protein Assay Kit, 5% of the lysate was taken as an input sample and supplemented with five-fold concentrated SDS-PAGE sample buffer, followed by heat denaturation at 95 °C. To account for the much stronger TurboID expression in cells expressing free TurboID in comparison to TurboID-TAX1BP1 expressing cells, 1 mg protein from the lysate of free TurboID expressing cells and 8 mg protein from the lysate of TurboID-TAX1BP1 expressing cells were incubated with 100 μl (free TurboID) or 250 μl (TurboID-TAX1BP1) pre-washed Streptavidin magnetic beads (three times washed with RIPA lysis buffer) overnight at 10 °C on a rotating wheel. On the next morning, beads were washed using the following washing buffers (1 ml per 100 μl beads each, room temperature) with the help of a magnetic rack: twice with RIPA lysis buffer (2 min), once with 1 M KCl (2 min), once with 0.1 M Na₂CO₃ (10 s), once with 2 M urea in 10 mM Tris-HCl (pH 8.0, 10 s), twice with RIPA lysis buffer (2 min), once with 1 M KCl (2 min), once with 0.1 M Na₂CO₃ (10 s), once with 2 M urea in 10 mM Tris-HCl (pH 8.0, 10 s) and twice with RIPA lysis buffer (2 min). Finally, 40–120 μl three-fold concentrated SDS sample buffer supplemented with 2 mM biotin and 20 mM DTT was added, and beads were boiled at 95 °C for 10 min. Eluates were transferred to a fresh tube, and 5%–10% of the eluate, as well as the input samples, were subjected to immunoblotting to check for successful enrichment of biotinylated proteins. The remaining eluates were stored for tryptic in-gel digestion and subsequent MS analysis. For the UBXN1 and PLAA proximitomics, the same protocol was applied, but 2 mg lysate from free TurboID expressing cells, 3 mg lysate from TurboID-UBXN1 expressing cells, and 5 mg lysate from PLAA-TurboID expressing cells were used and incubated with 200 μl streptavidin magnetic beads. TurboID proximity labeling experiments were performed in four (TurboID-TAX1BP1, PLAA-TurboID) or three (TurboID-UBXN1) biological replicates.

### Tryptic in-gel digestion

Approximately 90% of the eluates from free TurboID (control), TurboID-TAX1BP1, PLAA-TurboID or TurboID-UBXN1 samples were loaded onto 10% Tris-glycine wedge well gels, and proteins were allowed to migrate into the gel for 20 min at 150 V. Proteins were visualized using colloidal Coomassie Brilliant Blue and the protein-containing part of each gel lane was cut into 4 slices. Following destaining of the gel slices, cysteine residues were reduced with 5 mM Tris(2-carboxy-ethyl) phosphine dissolved in 10 mM ammonium bicarbonate (ABC) (incubation for 30 min at 37 °C), and free thiol groups were alkylated with 50 mM chloroacetamide/10 mM ABC (30 min at room temperature). Tryptic in-gel digestion using 0.06 μg of trypsin (sequencing-grade, Promega; dissolved in 10 mM ABC) per gel slice was performed overnight at 37 °C. The resulting peptides were then extracted through two rounds of incubation in a solution containing 0.05% (v/v) trifluoroacetic acid (TFA) and 50% (v/v) acetonitrile (ACN) in an ultrasonic bath for 10 min at 4 °C. Peptide-containing supernatants from each sample were combined. Peptides were subsequently dried under vacuum and desalted using StageTips[94]. In brief, StageTips were conditioned by sequentially adding methanol, 80% ACN in 0.5% acetic acid (v/v each), and 0.5% (v/v) acetic acid. Peptides were loaded onto StageTips, washed twice with 0.5% (v/v) acetic acid, and eluted with 80% ACN/0.5% acetic acid (v/v each). Solvents were evaporated, and dried peptides were stored at −80 °C until further use.

### Liquid chromatography-mass spectrometry analysis

Dried peptide mixtures were resuspended in 0.1% TFA and analyzed by nano-HPLC-ESI-MS/MS on an Orbitrap Elite (TurboID-TAX1BP1 and PLAA-TurboID experiments) or a Q Exactive Plus (TurboID-UBXN1 experiments) mass spectrometer (Thermo Fisher Scientific, Bremen, Germany), each coupled to an UltiMate 3000 RSLCnano HPLC system (Thermo Fisher Scientific, Dreieich, Germany). The RSLC systems were equipped with PepMap C18 precolumns (5 mm × 300 μm inner diameter, Thermo Scientific) and either a nanoEase M/Z HSS C18 T3 analytical column (length, 250 mm; inner diameter, 75 μm; particle size, 1.8 mm; packing density, 100 Å, Waters) for measurements at the Orbitrap Elite or an Acclaim PepMap analytical column (length, 50 cm; inner diameter, 75 μm; particle size, 2 μm; pore size, 100 Å, Thermo Scientific) for analyses at the Q Exactive. Peptides from TurboID-TAX1BP1 and PLAA-TurboID experiments were separated and eluted using a solvent system consisting of 0.1% (v/v) formic acid (FA) (solvent A) and 50% (v/v) methanol/30% (v/v) ACN/0.1% (v/v) (FA) (solvent B) and applying a gradient ranging from 1% to 7% solvent B in 5 min, 7% to 65% B in 65 min, and 65% to 80% B in 5 min at a flow rate of 300 nl/min. The solvent system for the analysis of peptides from TurboID-UBXN1 experiments consisted of 4% (v/v) dimethylsulfoxide (DMSO)/0.1% (v/v) FA (solvent A′) and 48% (v/v) methanol/30% (v/v) ACN/4% (v/v) DMSO/0.1% (v/v) FA (solvent B′) and the LC gradient was as follows: 1%–5% solvent B′ in 6 min, 5%–22% B′ in 100 min, 22%–42% B′ in 50 min, and 42%–80% B′ in 5 min; flow rate, 300 nl/min.

Peptides eluting from the analytical columns were transferred to a stainless steel emitter (Thermo Scientific; Orbitrap Elite) or a fused silica emitter (PicoTip, New Objectives; Q Exactive) for electrospray ionization using a Nanospray Flex ion source with DirectJunction adapter (Thermo Scientific) and applying a spray voltage of 1.8 kV (Orbitrap Elite) or 1.5 kV (Q Exactive) and a capillary temperature of 200 °C.

LC-MS/MS data were acquired in data-dependent mode applying the following parameters for measurements at the Orbitrap Elite: MS precursor scans at *m/z* 370 - 1700 with a resolution of 120,000 (at *m/z* 400); automatic gain control (AGC) set at 1 × 10⁶ ions; a maximum injection time (IT) of 200 ms; a TOP20 method for low-energy collision-induced dissociation of multiply charged precursor ions applying a normalized collision energy of 35%, an activation q of 0.25, an activation time of 10 ms, and an MS/MS isolation width of 2 Da. The AGC

for MS/MS scans was set at $5 \times 10^3$ ions with a maximum IT of 150 ms, and a dynamic exclusion time of 45 s was applied. Parameters for measurements at the Q Exactive were as follows: range for MS precursor scans, *m/z* 375 - 1700; resolution, 70,000 (at *m/z* 200); AGC, $3 \times 10^6$ ions; max fill time, 120 ms; TOP12 method for higher-energy collisional dissociation of multiply charged peptides; isolation width of 2 Da; AGC, $1 \times 10^6$; max fill time (orbitrap), 120 ms; normalized collision energy, 28%; dynamic exclusion time, 45 s.

## MS data analysis

MS/MS data were processed using the MaxQuant software package (version 2.4.4.0) (ref. [95]) with its integrated Andromeda search engine[96] and searched against the UniProt human reference proteome (proteome ID UP000005640; downloaded August 2023). Protein identification relied on the detection of at least one unique peptide with a minimum length of seven amino acids and a false discovery rate of 0.01 applied to both lists of peptide and protein identifications. 'Trypsin/P' was set as a proteolytic enzyme, and a maximum of three missed cleavages was allowed. Mass tolerances for precursor and fragment ions were 20 ppm and 0.5 Da, respectively. Carbamidomethylation of cysteine residues was set as fixed, N-terminal acetylation and methionine oxidation as variable modifications.

The autoprot Python module (v0.2) (ref. [97]) was used for further analysis of MaxQuant results and for data visualization. Label-free protein quantification was based on MS intensities. To facilitate the calculation of ratios in case a protein was not identified in a sample, missing values were imputed. First, intensity values missing in free TurboID replicates were imputed when values were present for all TurboID-TAX1BP1, PLAA-TurboID, and TurboID-UBXN1 replicates, respectively. Protein abundance ratios were then calculated from $\log_2$-transformed intensity values. Only proteins with ratios in two out of four (TurboID-TAX1BP1, PLAA-TurboID) or three (TurboID-UBXN1) biological replicates were considered for further analysis, and missing ratios were imputed. Proteins significantly enriched in TurboID-TAX1BP1, PLAA-TurboID, or TurboID-UBXN1 experiments were determined according to the "linear models for microarray data" (limma) method[98,99]. P-values were corrected for multiple testing following the Benjamini–Hochberg approach[100].

Information about proteins identified and quantified in the experiments is provided in Supplementary Data 1. A Jupyter notebook providing documentation of the analysis pipeline and statistical tools used is available at https://github.com/ag-warscheid/TAX1BP1_Proximitome and https://doi.org/10.5281/zenodo.12552577.

## Preparation of immunofluorescence samples

Cells grown on coverslips to 80% confluence were washed twice with PBS, fixed using 3.7% formaldehyde in PBS for 15 min at RT, washed twice with PBS, permeabilized with 0.2% Triton X-100 in PBS for 10 min at RT, washed with PBS, and blocked by incubation with 2% BSA in PBS for 30 min at RT. Cells were incubated with the indicated primary antibodies (diluted in 2% BSA in PBS) overnight at 4 °C, washed for 15 min with PBS, and incubated with the appropriate fluorophore-coupled secondary antibodies (diluted 1:500 in 2% BSA in PBS) for 2 h at RT. Hoechst staining was performed by incubating the cells for 10 min in PBS containing 2.5 µg/ml Hoechst 33342. Following this, cells were washed for 15 min with PBS and rinsed with water. Coverslips were mounted for microscopy with ProLong Glass Antifade Mountant.

## Fixed-cell microscopy

Confocal immunofluorescence microscopy was performed at the Imaging Core Facility (Biocenter, University of Würzburg) using Leica TCS SP2 or SP8 confocal microscopes equipped with an acousto optical beam splitter, or at the Core for Imaging Technology & Education (CITE) at Harvard Medical School, using a Yokogawa CSU-W1 spinning disk confocal on a Nikon Eclipse Ti-E motorized microscope.

At the SP2, single planes were acquired using a 63×/1.4 oil immersion objective, Diode UV (405 nm), Ar (488 nm), and HeNe (561 nm) lasers, and Leica software. At the SP8, single planes were acquired using a 63× glycerol objective lens, a white laser at 405 nm, 488 nm, 550 and 633 nm excitation, and Leica software.

At the Yokogawa CSU-W1 spinning disk confocal microscope, z-stacks were imaged with a Nikon Apochromat 60×/1.42 oil-objective lens. Signals of 405 nm, 488 nm and 568 nm fluorophores were excited in sequential manner with a Nikon LUN-F XL solid state laser combiner (laser power: 405 nm – 80 mW, 488 nm – 80 mW, 561 nm – 65 mW) using a Semrock Di01-T405/488/568/647 dichroic mirror. Fluorescence emissions were collected with Chroma ET455/50 m (for $\lambda_{ex}$ 405 nm), Chroma ET525/50 m (for $\lambda_{ex}$ 488 nm), and Chroma ET605/52 m (for $\lambda_{ex}$ 561 nm) filters, respectively (Chroma Technologies). Confocal images were acquired with a Hamamatsu ORCA-Fusion BT CMOS camera (6.5 µm² photodiode, 16-bit) and NIS-Elements image acquisition software.

To analyze larger cell numbers, immunofluorescence microscopy was performed on an Operetta High-Content Imaging System (Revvity, 20×/0.4, air, non-confocal, random field-of-view selection). Images were analyzed using Harmony High-Content Imaging and Analysis Software (Revvity).

Consistent laser intensity and exposure time were applied to all the samples, and brightness and contrast were adjusted equally by applying the same minimum and maximum display values in Fiji.

## 3D structured illumination microscopy

Fixed-cell 3D-SIM samples were prepared as described[101]. Briefly, HeLa^TMEM192-3xHA control and HeLa^TMEM192-3xHA *ATG7* KO cells were seeded on 18 × 18 mm Marienfeld Precision cover glasses, thickness No.1.5H (tol. ± 5 µm) and cultured at the indicated conditions/treatments. Cells were washed three times with PBS, and fixed and permeabilized with ice-cold methanol for 15 min. After three washes with 0.02% Tween 20 in PBS (PBST), cells were blocked for 10 min in 3% BSA in PBS at room temperature and washed again three times in PBST. Primary antibody incubation (rabbit anti-LC3 CST Cat# 3868, mouse anti-ubiquitin Enzo Cat# ENZ-ABS840-0500) was performed overnight at 4 °C with gentle rocking in 3% BSA in PBS, followed by three 5 min washes with PBST. Secondary antibody incubation (1:400 in 3% BSA in PBS) was performed at room temperature for 1 h with gentle rocking. To stain nuclei, DAPI (1:10,000) was added for 5 min to cells in PBST, and samples were washed three times for 5 min in PBST. Before mounting on glass slides, coverslips were washed once in PBS and mounted in Vectashield.

3D-SIM microscopy was performed on a DeltaVision OMX v4 using an Olympus 60×/1.42 Plan Apo oil-objective (Olympus, Japan) equipped with 405 nm, 445 nm, 488 nm, 514 nm, 568 nm, and 642 nm laser lines (all ≥100 mW). Images were recorded on a front-illuminated sCMOS (PCO Photonics, USA) in 95 MHz, 512x512px image size mode, 1× binning, 125 nm z-stepping with 15 raw images taken per z-plane (5 phase-shifts, 3 angles). Raw image data were computationally reconstructed using CUDA-accelerated 3D-SIM reconstruction code (https://github.com/scopetools/cudasirecon) based on ref. [102]. Optimal optical transfer function (OTF) was determined via a software developed in-house by Talley Lambert from the Core for Imaging Technology & Education/Cell Biology Microscopy Facility (https://github.com/tlambert03/otfsearch); all channels were registered to the 528 nm output channel, Wiener filter: 0.002, background: 90. ChimeraX was used for 3D renderings of 32-bit image stacks. After import, channels were visualized as surfaces by the Volume Viewer Tool, and surface presentations were cleaned up using the "Hide dust" command, based on size filtering thresholding.

## Image analysis

Image processing was performed using Fiji[103] and CellProfiler[104].

To quantify the percentage of cells with an aggresome and the total aggregate area per cell, AggreCount[105], an automated image analysis tool in Fiji, was adapted and applied unless otherwise mentioned. A minimum cutoff size of $5\,\mu m^2$ was applied for aggresomes, and a minimum cutoff size of $0.1\,\mu m^2$ was used for aggregates. Alternatively, to calculate the total aggregate area per cell, aggregates were identified in Fiji as regions by auto-thresholding (RenyiEntropy) the filtered and background-subtracted images. For the quantification of the mean aggregate area of the 3D-SIM images, a Gaussian filter was applied to the maximum intensity projections in CellProfiler. Aggregates were subsequently identified with the IdentifyPrimaryObjects command using a three-class adaptive thresholding strategy (Otsu), followed by a measurement of the area.

For quantification of colocalization between ubiquitin and LC3, ubiquitin and TAX1BP1, TAX1BP1 and WIPI2, or PFN2 and TAX1BP1, respectively, a custom-written Fiji batch macro was used that is available upon request. In brief, images of HeLa cells were subjected to background subtraction (rolling ball radius 7–15 pixels) after having applied a median-based filter of 1–2 pixels. Images were then auto-thresholded using the Renyi entropy option and merged, followed by a colocalization analysis using the ComDet plugin (v.0.5.5). Images of differentiated SH-SY5Y cells were also subjected to background subtraction (rolling ball radius 5 pixels), but in this case, the median-based filter was not applied. After auto-thresholding (RenyiEntropy) and merging the images, colocalization was analyzed in the same manner.

To quantify the p97 intensity in the nucleus and cytoplasm, as well as at the aggresome, a pipeline was created using CellProfiler to define nuclei via the Hoechst staining (thresholding method: minimum cross-entropy) and cells via the p97 staining (propagation from nuclei, thresholding method: otsu). The cytoplasm was defined by subtracting nuclei from cells. Following this segmentation, intensities were measured in the nucleus and the cytoplasm, and their ratio was calculated. Aggregates were detected with the IdentifyPrimaryObjects command, using a masked image of the ubiquitin or TDP-43 staining as input and an adaptive threshold strategy (robust background, variance method: three standard deviations). To select for aggresomes, aggregates were filtered by their area (minimum value 250). The p97 intensity was then measured at aggresomes and normalized to the intensity of the respective whole cell. This pipeline was also used to quantify the percentage of cells with TDP-43-positive aggresomes.

## Statistical testing

All statistical analyses and graphing were performed using GraphPad Prism 9. As described in the individual figure legends, Student's t-tests (two-tailed) or factorial ANOVAs (one- or two-way, corrected for multiple comparisons by Dunnett's or Bonferroni's multiple-comparisons test unless otherwise indicated) were applied to the raw data. A p-value $< 0.05$ was considered significant and indicated as follows: ns, not significant, $*p < 0.05$, $**p < 0.01$, $***p < 0.001$, $****p < 0.0001$.

## Reporting summary

Further information on research design is available in the Nature Portfolio Reporting Summary linked to this article.

# Data availability

Mass spectrometric raw data and MaxQuant result files have been deposited to the ProteomeXchange Consortium[106] via the PRIDE partner repository[107] and are accessible using the dataset identifier PXD053581. Source data are provided with this paper.

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

## Acknowledgements

We thank Martin Eilers, Mark Hochstrasser, Manuel Kaulich, Sascha Martens, Thomas U. Mayer, Magdalini Polymenidou, Alice Ting, Didier Trono, and Feng Shang for providing plasmids and cell lines; Petra Beli and Nadia da Silva Fernandes Lucas for advice on OtUBD pulldowns; the Imaging Core Facility (Biocenter, University of Würzburg) as well as the Core for Imaging Technology & Education and the Cell Biology Microscopy Facility (Harvard Medical School) for support with microscopy; Susanne Meyer, Carola Sommer and Romy Baier for excellent technical assistance. This work was funded by the Deutsche Forschungsgemeinschaft (DFG, German Research Foundation) through grants GRK2243/1+2 and BU951/5-1 (to A.B.), GRK2243/2, SPP2453 (project number 541758684; WA 1598/7-1) and FOR2743 (to B.W.), 440766788 (INST 93/1023-1-FUGG; Operetta CLS system), and HBFG-

133-612 (Leica SP2 confocal laser scanning microscope), as well as by the NIH (R01NS083524, R01NS110395 to J.W.H). This work has received funding from the European Union's Horizon 2020 research and innovation program under the Marie Skłodowska-Curie grant agreement No 812968 (H.D. and B.W.). F.K. was supported by the Goldberg Fellowship of the Cell Biology Education and Fellowship Fund at Harvard Medical School.

## Author contributions

M.K. and A.B. conceived the project; M.K., P.M., H.D., F.K., T.P., and C.S.-V. performed experiments; M.K., P.M., H.D., F.K., S.S., S. Oe., J.W.H., B.W., and A.B. analyzed the data; M.K., P.M., and A.B. wrote the manuscript.

## Funding

## Competing interests

J.W.H. is a consultant and founder of Caraway Therapeutics (a wholly owned subsidiary of Merck & Co., Inc.) and is a member of the scientific advisory board for Lyterian Therapeutics. The remaining authors declare no competing interests.
