## [Transparent Peer Review file · Nature Communications]

p97/VCP is required for piecemeal autophagy of aggresomes

Corresponding Author: Professor Alexander Buchberger

Version 0:

Reviewer comments:

Reviewer #1

(Remarks to the Author)

In this manuscript, the authors showed that aggresomes are cleared via selective autophagy requiring the cargo receptor TAX1BP1. In addition, they found that Hsp70 and p97/VCP with its cofactors UFD1-NPL4 and FAF1 play key roles in aggresome disassembly.

This study provides a potential pathway of autophagy-dependent aggresomes clearance. However, all experiments were performed in HeLa, which is a cancer cell line, making it questionable if this phenotype is general to all cells or only specific to some type of cancer cells. Therefore, for some key experiments to support the proposed conclusion, it's necessary to confirm it at least in another cell line.

In addition, the authors should show the biological output of these phenomena, for example, certain in vivo data are needed to demonstrate the importance of the p97/VCP-mediated aggresome clearance.

There are some other minor points that shall be taken care:

1. Fig3d's quality needs to be improved.
2. In most figures, the author used LC3 as a marker for staining autophagosome. LAMP2 or other lysosome markers should also be shown.

Reviewer #2

(Remarks to the Author)

Although aggresome clearance has been previously studied by many groups, most of these studies tend to over-express aggregation-prone proteins to promote the formation of aggresome. Here, Körner and colleagues have established a paradigm whereby HeLa cells were treated with the proteasome inhibitor, Bortezomib (Btz), that was subsequently washed out to promote the formation and (autophagic) clearance of endogenous aggresomes. Using this system, they identified TAX1BP1 as an essential autophagy cargo receptor. They further showed that Hsp70 and p97/VCP and its cofactors UFD1-NLP4 are involved in aggresome disintegration, which is important in presenting its components for piecemeal autophagy. Piecemeal autophagy is not a new concept, but the study conducted in this report is rather extensive and has illuminated some interesting players. Notwithstanding, I have several comments/suggestions for the author's consideration.

1. The aggresome formation/clearance paradigm involving a Btz wash-out period is interesting. As autophagy is known to proceed even in the presence of proteasome inhibition as a compensatory means of protein degradation when the proteasomal pathway is blocked, would the result be similar if there is no wash-out period? (As the authors are aware, under pathophysiological conditions, chronic proteasome dysfunction can occur, but cells can survive due to the presence of a functional autophagy system)

2. Pg 4 line 81, the authors have alluded to the finding by others that HDAC6 was found to be involved with the clearance of endogenous aggresomes. Is HDAC6 identified in the proximitome in this study? Does it play a role in the p97/UFD1/NLP4-mediated clearance of aggresome?

3. Fig. 1j: Would genetic re-introduction of TAX1BP1 rescue the clearance defects in TAX1BP1KO cells?

4. In Figures 3a-c, the addition of Btz resulted in no or less significant increase in the percentage of cells with aggresomes and total aggregate area per cell respectively compared to the addition of CB or VER. However, in Figure 3d, Btz significantly increased the accumulation of RIPA-insoluble aggregates, while VER had a more modest effect. Does this

suggest that Hsp70 activities may not play a major role in aggresome disaggregation, contrary to the conclusion in lines 167-169?

5. HOIP was identified by the authors to co-localize with aggresomes (Fig. 4a). Is HOIP found in the proximity? If not, what is the rationale of looking at HOIP? Interestingly, HOIP activity was found by the authors to be critical in the early proteotoxic stress response, but not during aggresome clearance, suggesting that linear M1-linked ubiquitin chain assembly is not needed for the removal of aggresomes. What about unanchored K63-linked ubiquitin chains, which has been shown in reference #33 (Hao et al., 2013) that it activates aggresome clearance?

6. Fig. 5a: Why is there a very strong signal detected by FAM83D in the absence of Btz and recovery (1st lane of pull-down)?

7. Finally, it will be interesting to see if the phenomenon is observed in neurons, as neuronal autophagy is known to be constitutive and unlike non-neuronal cells, neurons exhibit rapid autophagic flux. Moreover, most neurodegenerative diseases are characterized by the presence of protein aggregates.

Minor

- Please include in the materials and methods section on how total aggregate area per cell are quantified.
- Why was Baf added for 6 hours to inhibit autophagosome-lysosome fusion in Figure 1g, but only for 3 hours in the proximity proteomics assay? Was autophagosome-lysosome fusion effectively inhibited at 3 hours with Baf?
- Fig. 5e: Provide a quantification of total aggregate area per cell.
- Fig. 6: Does "AGM" refer to aggresome?
- Fig S2G: There seems to be little or no colocalization of PFN2 with TAXBP1. This is different from that observed in Fig S6A, which is curious.
- Any concern about off target effects of using a pharmacological strategy to inhibit the proteasome?

Reviewer #3

(Remarks to the Author)

In this manuscript, the authors reveal the important role of p97/VCP for aggrephagy. The experiments were performed with a high standard. The authors should address the following issues before publication.

1. The authors claim that "The p97 unfoldase and the Hsp70 system cooperate in aggresome disintegration". However, their data only support the concurrence of these two PQC systems. The blocking of Hsp70 by VER didn't show much difference compared to the case with CB for block p97. The collaboration between the two systems hasn't been demonstrated.
2. The SIM images shown in Figure 6 are very strange. Without a scale bar, people cannot estimate the size of their images. Moreover, the authors only show some representative images, which is not enough to demonstrate their conclusion. A quantitative analysis over ten cells for each condition is absolutely required.
3. The schematic illustration of Figure 7 is too simple. There is not much one can learn from it.

Version 1:

Reviewer comments:

Reviewer #1

(Remarks to the Author)

The authors have fully addressed my concerns in the revision. I would like to recommend it for acceptance.

Reviewer #2

(Remarks to the Author)

I am pleased to note the additional experiments that the authors have performed in their revised manuscript, particularly the requested genetic rescue experiment. These additional data provides further support to their conclusions. I am overall satisfied with the author's response to my comments.

Reviewer #3

(Remarks to the Author)

All my concerns have been addressed.

We thank all three Reviewers for their constructive and encouraging feedback. As detailed in the point-by-point response below, we addressed all their comments, many with additional experiments, to significantly improve the revised version of our manuscript.

Specifically, we

- confirmed the role of p97 in aggresome clearance in differentiated, neuron-like SH-SY5Y cells (new Fig. 4 and new Suppl. Fig. 5a, b)
- identified pleckstrin-2 (PLEK2) as a ubiquitylated aggresomal client that is partially cleared via aggrephagy (new Fig. 6a-g)
- determined the contributions of Poh1/RPN11/PSMD14 and HDAC6 to aggresome disintegration (new Suppl. Fig. 4a, b, f, g)
- demonstrated that similar kinetics of aggresome formation and clearance are observed upon addition of Btz, MG-132 and Epoxomicin, with or without inhibitor wash-out (new Suppl. Fig. 1b-d)
- added quantifications (new Fig. 6i, new Fig. 7k)
- improved the schematic depicting our model (new Fig. 8)

All author responses below and changes in the revised manuscript are highlighted by blue color.

Reviewer #1 (Remarks to the Author):

In this manuscript, the authors showed that aggresomes are cleared via selective autophagy requiring the cargo receptor TAX1BP1. In addition, they found that Hsp70 and p97/VCP with its cofactors UFD1-NPL4 and FAF1 play key roles in aggresome disassembly.

This study provides a potential pathway of autophagy-dependent aggresomes clearance. However, all experiments were performed in HeLa, which is a cancer cell line, making it questionable if this phenotype is general to all cells or only specific to some type of cancer cells. Therefore, for some key experiments to support the proposed conclusion, it's necessary to confirm it at least in another cell line.

Author response: The key finding of our study is that inhibition of p97/VCP completely blocks aggresome clearance by interfering with aggresome disintegration. Please note that we had shown this not only for HeLa cells, but also for A549 lung cancer cells and, importantly, for non-transformed human ARPE-19 cells and non-transformed murine C2C12 myoblasts (original Figs. 3e-g, S3h-j; now Fig. 4g-i and Suppl. Fig. 5c -f), clearly demonstrating that the phenotype is not specific to cancer cells. In the revised manuscript, we additionally confirmed the aggresome clearance defect in differentiated SH-SY5Y cells (new Fig. 4a-f, new Suppl. Fig. 5ab), which are an established in vitro model for studying aggresomes and other disease-linked protein aggregates in neuron-like cells (e.g. PMIDs 14645198, 33866701). Together, our data demonstrate that p97 is generally required for aggresome clearance in various cell types.

In addition, the authors should show the biological output of these phenomena, for example, certain *in vivo* data are needed to demonstrate the importance of the p97/VCP-mediated aggresome clearance.

Author response: Our study specifically addressed the role of p97/VCP during aggresome clearance by applying pharmacological p97/VCP inhibitors at the start of the clearance phase, after wash-out of the aggresome inducer Bortezomib.

Unfortunately, this approach cannot be taken with *in vivo* models, because these models lack the precise temporal and spatial control of the inhibitors' pharmacological effects that would be required to study aggresome clearance.

However, the following observations strongly support the importance of aggresome clearance and the role of p97/VCP *in vivo*:

- Aggresome-like proteinaceous inclusions can be induced in mouse brains by local pharmacological inhibition or genetic depletion of the 26S proteasome (PMIDs 18701681, 20649845).
- The intraneuronal inclusions in these mouse models resemble disease-linked Lewy bodies and cause neurodegeneration, thus linking failure to clear these inclusions to neurodegeneration. In fact, Lewy bodies share a number of features with aggresomes and have been proposed to resemble failed, persistent aggresomes (PMID 15261611).
- Missense mutations in the *VCP* gene cause multisystem proteinopathy 1 (MSP1), which is characterized by ubiquitin- and TDP-43-positive inclusions in neurons and muscle fibers, strongly suggesting that impaired p97-mediated clearance of aggregates/aggresomes contributes to MSP1 pathogenesis (PMID 21892620).

To underline the biomedical relevance of p97/VCP-mediated aggresome clearance, we included this information in lines 84ff., 332f., and 444ff. of the revised manuscript.

There are some other minor points that shall be taken care:

1. Fig3d's quality needs to be improved.

Author response: While we believe that the old Fig. 3d convincingly showed the accumulation of ubiquitylated proteins in the RIPA-insoluble fraction in the presence of all four inhibitors, we repeated the experiment and replaced the figure with an improved blot.

2. In most figures, the author used LC3 as a marker for staining autophagosome. LAMP2 or other lysosome markers should also be shown.

Author response: To distinguish between autophagosomes and (auto)lysosomes, we performed double immunostainings using LC3 and LAMP2 antibodies (see Figure for Reviewer R1a below). It is evident that at 15h recovery (peak of aggresome formation), +6h control (normal clearance) and +6h CB (persistent aggresomes), there is hardly any co-localization of LC3 and LAMP2, indicating that the quantification of auto/aggrephagosomes using LC3 as autophagosome marker is not significantly affected by any LC3-positive autolysosomes. Upon 6h Baf treatment, there is some overlap of LC3 and LAMP2 signals, but the massive perinuclear accumulation of signals prevents the reliable quantification of double-positive puncta. Similarly, in double stainings of TAX1BP1 and LAMP2, there is some overlap upon Baf treatment, but not under the other conditions (Fig. R1b). To account for this small

population of putative autolysosomes, we added "and autolysosomes" in the text when referring to LC3 as a marker in the presence of Baf. Importantly, the central conclusion that p97 inhibition interferes with aggrephagosome formation is not affected, because no overlap of LC3/TAX1BP1 and LAMP2 signals is observed under this condition.

Fig. R1: LC3- or TAX1BP1-positive puncta are rarely positive for LAMP2.

HeLa cells treated with Btz (1 μ M, 8h) were fixed after 15h recovery, or recovered for six additional hours in the absence or presence of 1 μ M CB-5083 or 100 nM BafA1, and subsequently analyzed by confocal immunofluorescence microscopy using antibodies against LC3 (a) or TAX1BP1 (b) and LAMP2. Scale bars, 50 μ m.

Reviewer #2 (Remarks to the Author):

Although aggresome clearance has been previously studied by many groups, most of these studies tend to over-express aggregation-prone proteins to promote the formation of aggresome. Here, Körner and colleagues have established a paradigm whereby HeLa cells were treated with the proteasome inhibitor, Bortezomib (Btz), that was subsequently washed out to promote the formation and (autophagic) clearance of endogenous aggresomes. Using this system, they identified TAX1BP1 as an essential autophagy cargo receptor. They further showed that Hsp70 and p97/VCP and its cofactors UFD1-NLP4 are involved in aggresome disintegration, which is important in presenting its components for piecemeal autophagy. Piecemeal autophagy is not a new concept, but the study conducted in this report is rather extensive and has illuminated some interesting players. Notwithstanding, I have several comments/suggestions for the author's consideration.

1. The aggresome formation/clearance paradigm involving a Btz wash-out period is interesting. As autophagy is known to proceed even in the presence of proteasome inhibition as a compensatory means of protein degradation when the proteasomal pathway is blocked, would the result be similar if there is no wash-out period? (As the authors are aware, under pathophysiological conditions, chronic proteasome dysfunction can occur, but cells can survive due to the presence of a functional autophagy system)

Author response: We addressed this point by analyzing aggresome formation and clearance in the continuous presence of the proteasome inhibitors Btz, MG-132 and Epoxomicin and found that the formation and clearance of aggresomes was similar to the experiments including wash-out (new Suppl. Fig. 1d), even though the prolonged treatment with Btz affected cell viability. Thus, the protocol including a wash-out period does not only allow for a more precise control of the start of aggresome clearance, but also avoids toxic effects of prolonged inhibitor treatment.

2. Pg 4 line 81, the authors have alluded to the finding by others that HDAC6 was found to be involved with the clearance of endogenous aggresomes. Is HDAC6 identified in the proximitome in this study? Does it play a role in the p97/UFD1/NLP4-mediated clearance of aggresome?

Author response: HDAC6 was not identified in the TAX1BP1 proximitome datasets, in agreement with the observation by Hao et al (2013) that HDAC6 dissociates from aggresomes during the wash-out of proteasome inhibitor. We used the HDAC6 inhibitors Tubacin and Tubastatin A to address the role of HDAC6 in aggresome clearance. Consistent with Hao et al, we found that HDAC6 inhibition caused a significant clearance defect which, however, was less pronounced than that caused by CB treatment (new Suppl. Fig. 4fg). To address a potential role of HDAC6 in p97-UFD1-NLP4-mediated clearance, we combined the HDAC6 inhibitors with CB. We were unable to observe an additive effect on aggresome disintegration, because CB treatment alone already caused a virtually complete block (new Suppl. Fig. 4f), precluding a conclusion with respect to a potential effect of HDAC6 inhibition on p97-mediated disintegration. However, we found that the total aggregate area (including smaller protein aggregates) during clearance was increased upon combined inhibition of HDAC6 and p97 (new Suppl. Fig. 4g). This could indicate that HDAC6

is also involved in the clearance of smaller aggregates not requiring p97, but other interpretations are possible.

3. Fig. 1j: Would genetic re-introduction of TAX1BP1 rescue the clearance defects in TAX1BP1KO cells?

Author response: We re-introduced TAX1BP1 into a TAX1BP1 KO clone by lentiviral transduction to show that the (sub-)endogenous expression of TAX1BP1 fully rescues the aggresome clearance defect of the KO cells (new Suppl. Fig. 2d-g).

4. In Figures 3a-c, the addition of Btz resulted in no or less significant increase in the percentage of cells with aggresomes and total aggregate area per cell respectively compared to the addition of CB or VER. However, in Figure 3d, Btz significantly increased the accumulation of RIPA-insoluble aggregates, while VER had a more modest effect. Does this suggest that Hsp70 activities may not play a major role in aggresome disaggregation, contrary to the conclusion in lines 167-169?

Author response: For the interpretation of the RIPA-insoluble fraction shown in Fig. 3d, it is important to compare the intensity of the Ub signal at the start of aggresome clearance (15h recovery) to the endpoint (21h). While a significant decrease was observed at 21h in the absence of inhibitors, the presence of VER resulted in a much stronger Ub signal, consistent with a major role of Hsp70 in aggresome disaggregation as shown in Figs. 3a-c.

The even stronger signal seen at 21h for Btz can be explained by the fact that a 6hr-treatment with Btz alone (without prior aggresome formation) already induces the formation of RIPA-insoluble, ubiquitylated aggregates (original Suppl. Fig. S3c; now Suppl. Fig. 4e), in contrast to the other inhibitors. This additional RIPA-insoluble material seen upon addition of Btz likely reflects the de novo formation of small, mostly submicroscopic peripheral aggregates. With respect to aggresome clearance, the spatial resolution of the microscopy experiments shown in Figs. 3a-c overcomes this complication, allowing us to maintain our conclusions about the differential contributions of the PQC systems.

5. HOIP was identified by the authors to co-localize with aggresomes (Fig. 4a). Is HOIP found in the proximitome? If not, what is the rationale of looking at HOIP? Interestingly, HOIP activity was found by the authors to be critical in the early proteotoxic stress response, but not during aggresome clearance, suggesting that linear M1-linked ubiquitin chain assembly is not needed for the removal of aggresomes. What about unanchored K63-linked ubiquitin chains, which has been shown in reference #33 (Hao et al., 2013) that it activates aggresome clearance?

Author response: HOIP was not enriched in the TAX1BP1 proximitome datasets. The rationale for testing the involvement of HOIP were two previous publications by the Winklhofer lab, showing that HOIP promotes the M1 decoration and efficient clearance of large aggregates formed by disease-associated proteins such as polyQ-expanded exon 1 from Huntingtin, polyQ-expanded Ataxin-3, alpha-synuclein, and mutant SOD1 and TDP-43 (PMIDs 30886048, 38114471). We slightly rephrased lines 246ff. to make this clear.

Hao et al. published in 2013 that the proteasomal DUB Poh1 (now better known as RPN11 and PSMD14) generates unanchored K63-linked Ub chains, which in turn

activate HDAC6 to promote the formation of an actin network around aggresomes as a prerequisite for their efficient clearance. To address the role of Poh1/RPN11 in our experimental setting, we added the specific Poh1/RPN11 inhibitor Capzimin (PMID 28244987), which was not available in 2013, during aggresome clearance (new Suppl. Fig. 4ab). However, we did not observe any defect in aggresome disintegration. Compared to the DMSO control, a moderately elevated total aggregate area indicated the accumulation of smaller aggregates, similar to the effect of Btz. Our data thus indicate that the actual disintegration of aggresomes does not require the activity of Poh1/RPN11 and the 26S proteasome in our experimental setting. The unanchored K63-linked Ub chains reported to activate HDAC6 by Hao et al may be produced by Poh1/RPN11 during aggresome formation, or by a Poh1/RPN11-independent mechanism. We speculate that the strong aggresome clearance defect of Poh1/RPN11 knockdown cells observed by Hao et al may be an indirect consequence of altered cellular proteostasis due to prolonged genetic depletion of Poh1/RPN11, which is required for efficient proteasomal degradation (PMIDs 12183636, 12353037). While we agree with Hao et al that HDAC6 activity is required for efficient aggresome clearance (see point 2 above), we refrained from addressing the source and function of unanchored K63-linked Ub chains in more detail. Such analysis would be experimentally extremely challenging, partially redundant with the publication by Hao et al, and beyond the scope of the present study.

6. Fig. 5a: Why is there a very strong signal detected by FAM83D in the absence of Btz and recovery (1st lane of pulldown)?

Author response: We apologize for the bad quality of this blot - the strong signal was probably a blotting artifact. Our lab recently moved away from GST-TUBEs for pulldown experiments and instead prefers to use the ubiquitin binding domain derived from an *Orientia tsutsugamushi* deubiquitylase (OtUBD; PMID 35771886) for the enrichment of ubiquitylated proteins. OtUBD binds with very high affinity to various types of Ub conjugates, even under denaturing conditions, and was covalently linked to a resin, preventing its "bleeding" into the final sample. Using this superior approach, we confirmed the ubiquitylation of AMOTL2 (new Fig. 6a), but could not confirm the ubiquitylation of FAM83D (new Suppl. Fig. 7k). However, during our continued efforts to validate candidates from the TAX1BP1 proximitomes, we characterized PLEK2 (pleckstrin-2) as a new, ubiquitylated aggresomal client that is partially degraded via aggrephagy. We re-arranged the manuscript accordingly, moving the FAM83D data to new Suppl. Fig. 7h-k and including the PLEK2 data in new Fig. 6a-e and new Suppl. Fig. 3g.

7. Finally, it will be interesting to see if the phenomenon is observed in neurons, as neuronal autophagy is known to be constitutive and unlike non-neuronal cells, neurons exhibit rapid autophagic flux. Moreover, most neurodegenerative diseases are characterized by the presence of protein aggregates.

Author response: We thank the Reviewer for this excellent suggestion. Unfortunately, our attempts to induce aggresomes in iNeurons failed due to their high sensitivity to proteasome inhibitors. Instead, we analyzed aggresome clearance in differentiated SH-SY5Y cells (new Fig. 4a-f, new Suppl. Fig. 5ab), which are an established in vitro model for studying aggresomes and other neurodegenerative disease-linked protein aggregates in neuron-like cells (e.g. PMIDs 14645198, 33866701). Our data clearly

demonstrate that p97 is also required for normal aggresome clearance in neuron-like cells and that p97 inhibition induces the persistence of large, intact aggresomes.

Minor

- Please include in the materials and methods section on how total aggregate area per cell are quantified.

Author response: Done (lines 887ff.).

- Why was Baf added for 6 hours to inhibit autophagosome-lysosome fusion in Figure 1g, but only for 3 hours in the proximity proteomics assay? Was autophagosome-lysosome fusion effectively inhibited at 3 hours with Baf?

Author response: 3 hours of Baf treatment are fully sufficient to completely block autophagosome-lysosome fusion. The Baf treatment was extended to 6 hours to match the incubation time with that for the other inhibitors (which was chronologically established after the proximity proteomics had been performed).

- Fig. 5e: Provide a quantification of total aggregate area per cell.

Author response: A quantification of the total TDP-43-positive aggregate area per cell was added (new Fig. 6i).

- Fig. 6: Does “AGM” refer to aggresome?

Author response: Yes. The abbreviation is now defined in the figure legend.

- Fig S2G: There seems to be little or no colocalization of PFN2 with TAXBP1. This is different from that observed in Fig S6A, which is curious.

Author response: It is true that TAX1BP1 positive, mature aggresomes (as those highlighted in the zoom inset of original Fig. S2g) are hardly decorated with PFN2. By contrast, a strong co-localization of both proteins is observed at disassembling/disassembled aggresomes, suggesting a potential role in actin-mediated aggrephagosome movement. We set a new zoom window in original Fig. S2g (new Suppl. Fig. 3g) and updated original Fig. S6a (new Suppl. Fig. 8a) to better highlight the co-localization of TAX1BP1 and PFN2 at disassembled aggresomes, and we slightly re-phrased the text accordingly (lines 354 & 357).

- Any concern about off target effects of using a pharmacological strategy to inhibit the proteasome?

Author response: Since the Btz treatment to induce aggresomes was followed by a wash-out period of fifteen hours before the actual experiments were started, we consider it unlikely that the observed phenomena were caused by off-target effects of Btz. Nevertheless, we compared aggresome formation and clearance using three proteasome inhibitors representing distinct chemical classes (i.e., Btz, MG-132, Epoxomicin) and found that all three inhibitors induced aggresome formation and clearance, albeit with slightly different kinetics (Fig. 1a and new Suppl. Fig. 1b,c).

Moreover, for the experiments with SH-SY5Y cells (see point 7 above), we used MG-132 to induce aggresomes and found that addition of CB during clearance caused the same clearance defect as observed in our other experiments with Btz-induced aggresomes. Together, the high consistency between all our experiments, independent of the proteasome inhibitor used for aggresome induction, makes a significant contribution of any off-target effect to the observed phenotypes extremely unlikely.

Reviewer #3 (Remarks to the Author):

In this manuscript, the authors reveal the important role of p97/VCP for aggrephagy. The experiments were performed with a high standard. The authors should address the following issues before publication.

1. The authors claim that “The p97 unfoldase and the Hsp70 system cooperate in aggresome disintegration”. However, their data only support the concurrence of these two PQC systems. The blocking of Hsp70 by VER didn’t show much difference compared to the case with CB for block p97. The collaboration between the two systems hasn’t been demonstrated.

Author response: We agree with the Reviewer and apologize for the overstatement. Since CB alone causes a virtually complete block of aggresome disintegration (Fig. 3b), we were unable to show any further impairment upon combined treatment with CB and VER (original Suppl. Fig. S3d, now Suppl. Fig. 4h). A potential collaboration between the two systems was suggested by the effect of DNAJB6 depletion, which in combination with CB significantly increased the percentage of cells with persistent aggresomes (original Fig. 3hi, now Fig. 3ef). However, alternative interpretations of this result are possible. We therefore changed the wording to "The p97 and Hsp70 disaggregates *mediate* aggresome disintegration". Moreover, we stated in the Discussion section that it "remains to be determined if p97 and Hsp70 act in this process cooperatively on the same substrate proteins ... or if they act in parallel on distinct substrates." (lines 433ff.)

2. The SIM images shown in Figure 6 are very strange. Without a scale bar, people cannot estimate the size of their images. Moreover, the authors only show some representative images, which is not enough to demonstrate their conclusion. A quantitative analysis over ten cells for each condition is absolutely required.

Author response: We added scale bars to the images. We also quantified the mean area of the ubiquitin foci from a total of 29 - 36 cells per condition (new Fig. 7k), which confirms the significant accumulation of ubiquitylated structures in the presence of CB. A meaningful quantification of the LC3 decoration at ubiquitylated structures is not possible, because under the 6h control and 6h Baf conditions most aggresomes are disintegrated and sequestered into aggrephagosomes, where the presence of LC3 is trivial. However, the poor LC3 decoration of persistent aggresomes in the presence of CB can be readily appreciated qualitatively for the maximum intensity projections from a total of 12 cells per condition shown in original Suppl. Fig. S6f, now Suppl. Fig. 8f), which were all imaged with the same gain and identically processed.

3. The schematic illustration of Figure 7 is too simple. There is not much one can learn from it.

Author response: We improved the schematic (new Fig. 8) by putting a stronger emphasis on the requirement of p97-mediated disintegration for phagophore maturation/closure.